# Wind Speed and Direction Estimation from Wave Spectra using Deep Learning

Haoyu Jiang[1,2,3]

[1] Hubei Key Laboratory of Marine Geological Resources, China University of Geosciences, Wuhan, 430000, China
[2] Laboratory for Regional Oceanography and Numerical Modeling, Pilot Qingdao National Laboratory for Marine Science and Technology, Qingdao, 266000, China
[3] Southern Marine Science and Engineering Guangdong Laboratory (Guangzhou), Guangzhou 511458, China

*Correspondence to*: Haoyu Jiang (Haoyujiang@cug.edu.cn)

**Abstract.** High-frequency parts of ocean wave spectra are strongly coupled to the local wind. Measurements of ocean wave spectra can be used to estimate sea surface winds. In this study, two deep neural networks (DNNs) were used to estimate the wind speed and direction from the first five Fourier coefficients from buoys. The DNNs were trained by wind and wave measurements from more than 100 meteorological buoys during 2014-2018. It is found that the wave measurements can best represent the wind information about 40 minutes ago, because the high-frequency portion of the wave spectrum integrates preceding wind conditions. The overall root-mean-square error (RMSE) of estimated wind speed is ~1.1 m/s, and the RMSE of wind direction is ~14 ° when wind speed is 7~25 m/s. This model can not only be used for the wind estimation for compact wave buoys but also for the quality control of wind and wave measurements from meteorological buoys.

## 1 Introduction

Sea surface wind and waves are important parameters for the marine environment and ocean dynamics. High-quality simultaneous measurements of sea surface wind and wave information are helpful for the study of many oceanic and coastal phenomena. Such simultaneous measurements can be obtained from meteorological buoys and remote sensing satellites. Many meteorological buoys can provide comprehensive wind and wave information, such as surface wind speeds, wind directions, and wave spectra, with high accuracy. However, the deployment and maintenance of these buoys and platforms usually need relatively high costs. Therefore, meteorological buoys are very sparsely distributed and are mostly only available along the coastlines of developed countries.

The earth observation satellite network, such as scatterometers, altimeters, and synthetic aperture radars can serve as effective complements for the buoy network. Meanwhile, these remote sensors also have some limitations. Scatterometers can retrieve both wind speed and direction with a wide swath and the best overall accuracy, but wave information is not available from them. Besides, their temporal resolutions are (usually one or two revisits per day except for Polar Regions) still much lower than in-situ measurements. Altimeters can simultaneously measure wind speed and significant wave height (SWH), but wind directions and other wave parameters are not available from them. Besides, the cross-track spatial coverage and temporal

resolution of an altimeter are low because they can only measure the nadir. Synthetic aperture radars' wave mode can provide wind speed, wind direction, SWH, and low-frequency wave spectra (high-frequency is not available due to nonlinear imaging), but the accuracy of wind speed, wind direction, and SWH is usually not as good as those from scatterometers and altimeters,

and they are also limited by the sparse sampling. Moreover, space-borne remote sensors often perform worse in nearshore regions than in the open ocean due to the land contamination of backscatter.

Another important data source for collocated winds and waves is compact wave buoys. These types of buoys are usually low-cost and are suited for deploying in large numbers, and they perform better in measuring waves compared to large meteorological buoys because their small sizes have a more sensitive response to short waves (Voermans et al. 2020). Although

wave buoys are not designed for wind observation, Voermans et al. (2020) have shown that both wind speed and direction can be estimated from the wave spectra using a $f-4$ spectral dependence in the equilibrium range. Their model can estimate wind speed with a root-mean-square error (RMSE) of 2 m/s and wind directions with an RMSE of ~20 ° when wind speed is higher than 10 m/s. Although this model has good theoretical support, the accuracy of this model is lower than typical remote sensing retrievals. For example, altimeter-retrieved wind speed has a typical overall RMSE of 1.2-1.5 m/s (e.g., Jiang et al. 2020) and

scatterometer-retrieved wind speed and wind directions has a typical overall RMSE of ~1 m/s and 15 ° (e.g., Wang et al. 2021) when using buoys' anemometer data as the reference.

Compact wave buoys are increasingly widely used in global wave observations. For example, more than 2,000 Spotter buoys have been deployed in global oceans by Sofar Ocean Technologies (The location of these buoys can be viewed at https://weather.sofarocean.com/) to improve the performance of their wave modelling (Smit et al, 2021). Although the data is

not open to the public, more accurate wind estimation from wave spectra can definitely benefit users of such buoys. Voermans et al. (2020) have shown the possibility to estimate wind speed and wind direction with wave measurements alone. This study aims to improve the accuracy of such estimation as much as possible. A model based on a deep neural network (DNN) is presented to achieve this goal. The rest of this paper is organized as follows: The simultaneous observations of wind and waves to train the DNN model are introduced in Section 2, along with the structure and training method of the DNN. The main results

are presented in Section 3. A brief discussion about the selection of the DNN input terms is made in Section 4, followed by the concluding remarks in Section 5.

## 2 Data and Methods

### 2.1 Collocated Wind and Wave Data

Many buoys from the National Data Buoy Center (NDBC) coastal-marine automated network can provide quality-

controlled in-situ wave and wind measurements. The data used in this study is the NDBC buoy data archived in National Centers for Environmental Information where the data is available in NetCDF form. After removing the data records with bad-quality flags, more than 1.6 million records from 101 buoys in coastal and oceanic regions during 2014-2018 were used in this study (Fig. 1). Most buoys' anemometers are 4-5 meters from the sea surface, and winds are measured every ten minutes with sampling time of eight minutes and accuracy within 1 m/s and 10 ° for wind speed and direction, respectively, in moderate sea

state (in extreme sea states, the swing and tilting of the buoy can introduce larger errors). The wind speed was converted to the standard height of 10 m (U10) using the power law (Hsu et al. 1994) that was also used in Voermans et al. (2020). This conversion was also tried using the log profile (Young 1995), which has almost no impact on the results. The waves are measured every one hour with sampling time of 20 minutes. The buoy wave data includes five Fourier coefficients of waves for different frequencies in the range of 0.02-0.485 Hz (47 frequency bins) derived from the translational or pitch-roll

information from the accelerometers and inclinometers on board buoys (Steele et al. 1998). The five Fourier coefficients are wave variance spectral densities ($E$) which describe the wave energy for each frequency, mean and principal wave directions for each frequency ($\alpha_1$ and $\alpha_2$), and first and second normalized polar coordinates of Fourier coefficients ($r_1$ and $r_2$) which describe the directional spreading about the main direction the for each frequency. The five Fourier coefficients of different frequencies are the minimum requirement to reconstruct the directional wave spectrum. These NDBC data, especially the

offshore ones, are widely used in the validation of wind and wave remote sensing and numerical weather and wave models (e.g., Jiang et al. 2016, Jiang 2020, Wang et al. 2021). The wave data and the wind data were collocated if their ends of sampling time are within ten minutes (the sampling duration is ~20 minutes for wave measurements and ~10 minutes for wind measurement).

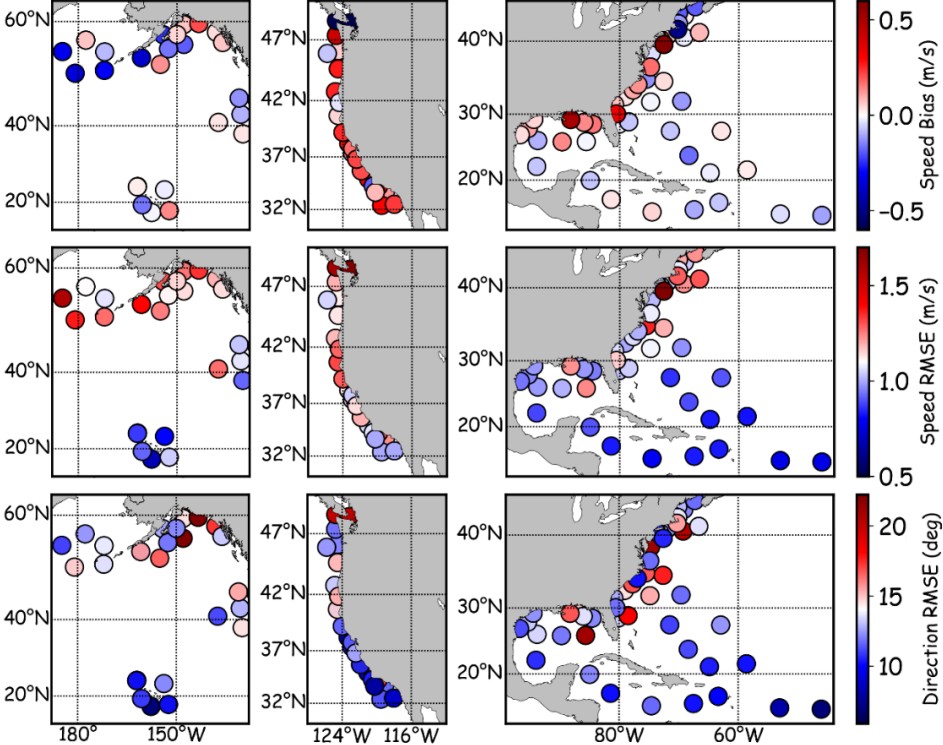

**Figure 1. The bias (1st row) and RMSE (2nd row) of DNN-estimated wind speed and RMSE of DNN-estimated wind direction (when wind speed is higher than 7 m/s, 3rd row) for the individual NDBC buoys in the North Pacific (left), the west coast of the United States (middle), and the Atlantic region (right). The overall RMSEs of wind speed and wind direction (when wind speed is higher than 7 m/s) are ~1.1 m/s and ~14°, respectively, for the complete validation data set. Therefore, blue and red colors in RMSE maps indicate below and above the overall RMSE, respectively.**

## 2.2 DNN Models for Estimating Wind Speed and Direction

As a nonparametric model, a DNN can theoretically be used to fit any form of function with any number of input parameters provided the network is wide and deep enough. The DNN has been proved to be effective for regression problems with more than two input parameters and is widely used in the training of retrieval models and correction models in studies of ocean remote sensing (e.g., Wang et al. 2020, Jiang et al. 2020). A DNN is a useful tool for the problem that there is causal relationships between inputs and outputs (in this study, wave spectra and winds, respectively) but the explicit form of the relationship is not known. In this study, two DNNs were established with the same structure, one for estimating wind speed and one for wind directions. In the beginning, the input layer of the DNN was set up in a "violent" way which simply contains 235 (vectorization of five Fourier coefficients $\times 47$ frequency bins) neurons. However, we will show in Section 4 that the input layer of the DNNs can be refined after obtaining the basic knowledge of how these models work. Each of the 235 inputs was normalized to have zero mean and unit variance. The DNNs have two hidden layers with 64 neurons followed by an output layer with one term (wind speed or direction). The activation function is the rectified linear unit (ReLU). It was tested that adding hidden layers and hidden neurons does not improve the performance of these models. The 1.7 million buoy records were randomly divided into training (50%) and validation (50%) sets. The DNN for U10 was trained to minimize the RMSE between the target (buoy-measured) and output U10:

$$Loss_{U10} = RMSE = \sqrt{\frac{1}{n}\sum_{i=1}^{n}(y_i - x_i)^2} \tag{1}$$

where $y$ and $x$ denote the output and target/reference parameters, respectively. The DNN for wind directions was trained to minimize the distance between target and output unit vector corresponding to the wind direction:

$$Loss_{Dir} = \sqrt{\frac{1}{n}\sum_{i=1}^{n}\left[(\sin(y_i) - \sin(x_i))^2 + (\cos(y_i) - \cos(x_i))^2\right]} \tag{2}$$

For both DNNs, the training used the Adam optimizer with a batch size of 2048. The learning rate (initially set to 0.004) was decreased by 50% if the loss of the training set did not decrease for two epochs, and the training process stopped when the RMSE of the validation set did not decrease for six epochs. The DNN was realized by PyTorch. Besides RMSE, the bias, STandard Deviation (STD), and Correlation Coefficient (CC) were also selected as the error metrics to evaluate the model performance:

$$Bias = \frac{1}{n}\sum_{i=1}^{n}(y_i - x_i) \tag{3}$$

$$STD = \sqrt{RMSE^2 - Bias^2} \tag{4}$$

$$CC = \sum_{i=1}^{n}(y_i - \bar{y})(x_i - \bar{x}) / \left[\sqrt{\sum_{i=1}^{n}(y_i - \bar{y})^2}\sqrt{\sum_{i=1}^{n}(x_i - \bar{x})^2}\right] \tag{5}$$

## 3 Results

The comparison between the collocated DNN-estimated and direct-measured U10 for the validation data set is shown as a scatterplot in Fig. 2a, and the corresponding comparison for wind directions is shown in Fig. 2d. These results suggest that estimating wind speed and direction from wave spectra using such a simple DNN works reasonably well. For wind speed, the DNN can give an estimation with an overall RMSE of ~1.3 m/s and a small overall bias. For wind direction, the RMSE is ~16 ° for U10 > 7 m/s. These results have significant improvement compared to the error metrics of Voermans et al. (2020).

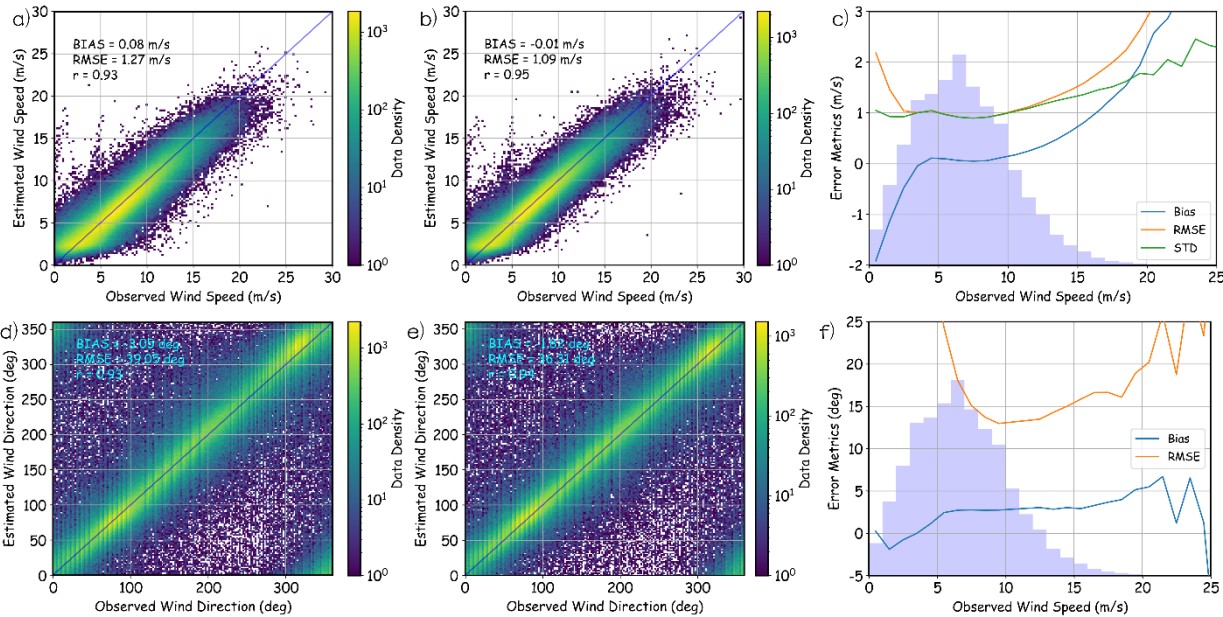

**Figure 2. (a-c) Comparison between wind speeds measured by buoys and those estimated by wave spectra. (a) Scatter plot of collocated DNN-estimated wind speed and direct-measured wind speed. (b) The same as (a), but the spectra were used to estimate the wind speed 40 minutes ago. (c) The bias, STD, and RMSE of the DNN-estimated wind speed one hour ago as a function of direct-measured wind speed. The blue shadow indicates the empirical distribution function of direct-measured wind speed. (d-f) The same as (a-c), but for wind directions.**

It is noted that the sampling duration is ~20 minutes for wave measurements and ~10 minutes for wind measurement. Different from the capillary waves with very high frequencies always in instant equilibrium with the local wind, the growth of gravity waves is time-dependent. Besides the current wind information, the wave spectrum measured by a buoy at a given location and time also contains remote and past wind information (Jiang and Mu 2019), because the wave spectrum is, to some degree, integrated winds. Therefore, it is possible that the buoy wave spectrum can better represent the local wind information some time ago. Based on this idea, the wave spectra were also collocated with past wind measurements using different time lags. For the collocations of each time lag, DNNs were re-trained to estimate the corresponding wind speed and directions and the error metrics were re-computed. The error metrics as a function of time lag were shown in Fig. 3. The results indicate that the DNN perform significantly better in estimating wind information a short period ago than the current wind information. The best error metrics for wind speed and wind direction were found at 40-50 minutes and 40-60 minutes before the end of wave

sampling time, respectively. Voermans et al. (2020) found that the wind acceleration is related to model error residuals, which is consistent with the results here.

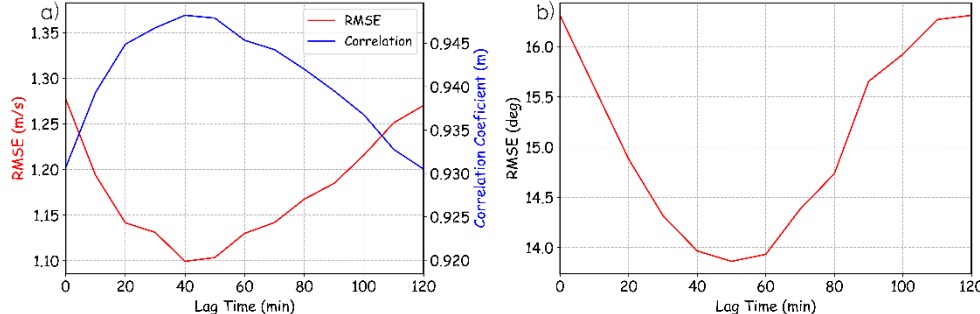


**Figure 3. (a) The RMSE and CC of the DNN-estimated wind speed as a function of lag time between wave and wind measurements (waves' end sampling time minus winds' end sampling time). (b) The RMSE of DNN-estimated wind direction as a function of lag time between wave and wind measurements for wind speed higher than 7 m/s.**

Obtaining wind information with only a 40-minute delay (near real-time) is acceptable for most scientific and operational applications. Therefore, the DNNs for wind of 40-minute delay were used in the following analysis. The results of wind speed and direction in the validation data set are shown in Figs. 2b and 2e, respectively. The corresponding error metrics as a function of direct-measured U10 are shown in Figs. 2c and 2f. The overall RMSE for U10 is ~1.1 m/s and is only ~1 m/s for U10 between 2 and 10 m/s where the sample size is relatively large. The DNN model tends to overestimate the U10 when it is lower

than 2 m/s, and the DNN model seldom gives an output of U10 less than 1 m/s. These are probably because the NDBC buoys do not well response to the small waves generated by very low wind while the geophysical noises such as ocean currents have a large impact on the wind-estimation during low wind speed. Meanwhile, it is noted that other indirect methods for wind speed estimation, such as remote sensing, also always overestimate low wind speed (e.g., Stopa et al. 2017, Jiang et al. 2020). Both the bias and STD increase with the U10 when U10 > 10 m/s. This is partly because the distribution of wind speed is not

uniform and the error of DNN is often larger for the less sampled conditions. Although the DNN model tends to underestimate high wind speed, the relative RMSE remains less than 14% for U10 < 20 m/s and the accuracy is also improved for high U10 compared to Voermans et al. (2020). For U10 > 20 m/s, the bias becomes higher than the STD, which means the systematic error becomes the main contributor to the RMSE. This is not surprising because the air-sea interaction becomes much more complicated during extreme wind and it is also noted that the U10 extrapolated from the wind speed measured at 4-5 m might

be overestimated to some extent in extreme sea states because the anemometers might be within the wave boundary layer (Babanin et al., 2018). The overall RMSEs of U10 retrieved from space-borne altimeters and scatterometers using corresponding state-of-the-art combinations of sensors and algorithms are ~1.2 m/s and ~1.0 m/s, respectively, compared to buoy-measurements (Jiang et al. 2020; Wang et al. 2021). According to the RSME, the accuracy of the DNN-estimated U10 is higher than altimeter U10 retrievals, and similar to scatterometer U10 retrievals if the data of U10 < 2 m/s is excluded.

For wind directions, the RMSE is larger than 25° when U10 < 5 m/s but decreases fast with the increase of U10. The RMSE becomes less than 20°, 15°, and 13° for U10 = 6, 8,10 m/s, respectively. Beyond U10 = 10 m/s, the RMSE of DNN-estimated wind directions slightly increases with the increase of U10 but remains < 20° until U10 > 21 m/s. It is noted that there were only less than 100 samples for U10 > 21 m/s, and most of them correspond to some strong cyclones where the directions of the wind vary rapidly. Following Voermans et al. (2020), if only the condition of U10 > 7 m/s was considered,

the overall RMSE of the DNN-estimated wind directions was only ~14°. To test the robustness of the DNN framework, we tried the random division, training, and validation processes more than 20 times, and the resulting error metrics in the validation data set stayed stable that there was no change in the first two significant digits of RMSEs of both U10 (1.1 m/s) and wind directions (14°). Wind direction information is also available from space-borne scatterometers, and the RMSE of wind directions between scatterometers (e.g., ASCAT-B/C, OSCAT2, HSCAT-B) and buoys is 15~18° according to Wang et al.

(2021). Therefore, the performance of the DNN model is also as good as state-of-the-art scatterometers with respect to wind directions for U10 > 7 m/s.

        The error metrics of the DNN-estimated wind information (with a time lag of 40 minutes) for different buoy locations are shown in Fig. 1. The error metrics vary with buoy locations. The distribution of U10 RMSE for individual buoys is similar to that of Voermans et al. (2020), but the RMSE values are much lower here. For most buoys in the open oceans to the South of

40°N, the RMSEs of DNN-estimated U10 and wind directions (for U10 > 7 m/s) are less than 1.0 m/s and 10°, respectively. Two buoys are found to have a U10 RMSE larger than 2 m/s: Station 44066 (2.1 m/s) at ~40°N in the U.S. East Coast and Station 46070 (2.2 m/s) in the southwest Bering Sea. It is noted that the biases of U10 for the two buoys (44066 and 46070) are also large. After a further check of the time series of measured and estimated U10, it is found that there seems to be an anemometer problem at Station 44066 from 22-Jan-2014 to 13-Feb-2014 (Fig. 4a). The measured and estimated U10 have a

good agreement before 22-Jan-2014, but the measured U10 values become significantly lower than the estimated ones after 22-Jan-2014. After a sudden drop on 26-Jan-2014, the measured U10 remains lower than 5 m/s for more than 15 days, which is unrealistic. A similar condition happened at Station 46070 from 03-Mar-2016 to 20-Apr-2016 (Fig. 4b), when the estimated U10 suddenly becomes significantly lower than the measured U10. Because the DNN model is unbiased and time-independent, such a systematic underestimation or overestimation of U10 for a long period has to be attributed to the problem of either wind

or wave sensor. Therefore, such a DNN-based U10 estimation model can also serve as an additional quality control/monitoring method for wind and wave sensors on meteorological buoys. If the bias between estimated and measured U10 remains significant for a short period (e.g., 3~5 days), the wind and wave data then needs to be further checked or discarded. Because the buoy data has been quality controlled by NDBC, such conditions were only identified in the two cases in Fig. 4. If we remove the bad-quality data in Fig. 4, the U10 RMSEs for Station 44066 and 46070 will drop to only 1.10 m/s and 1.25 m/s,

respectively.

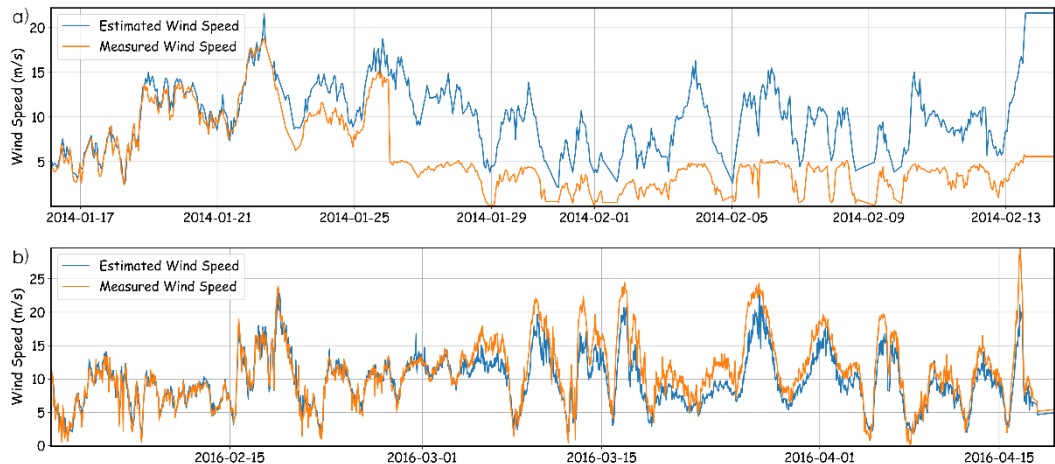

**Figure 4. Time-series comparison of direct-measured (orange) and DNN-estimate (blue) wind speed for (a) Station 44066 from 16-Jan-2014 to 15-Feb-2014 and (b) Station 46070 from 01-Feb-2016 to 20-Apr-2016. For 44066, the measured wind speed values became significantly lower than the DNN-estimated ones after 22-Jan-2014. For 46070, the DNN-estimated wind speed values became significantly lower than the direct-measured ones after 03-Mar-2016.**

The other two buoys with relatively high U10 RMSE (> 1.5 m/s), Station 46087 and 46088, are both at the Strait of Juan de Fuca where tidal currents are strong. First of all, the wind estimated from wave measurements is the wind relative to currents because waves are forced by relative wind. A strong current will make the estimated relative wind deviate from the absolute wind from the anemometer, introducing errors to the DNN model. Secondly, the phase velocity of the high-frequency waves and the current velocity are at the same order of magnitude during strong currents. In this case, the dispersion relation of high-frequency waves is strongly distorted by the currents via Doppler shift. This will lead to different frequency spectra for the same wavenumber spectra, introducing another error source for DNN-estimated wind speed. The surface currents are generally larger in coastal regions (tides) and westerlies (wind drifts) than in low-latitude open oceans, which can explain the spatial distributions of the U10 RMSE and can also partly explain why this model tends to underestimate large winds. Strong drifts along the wind direction will shift the wind-wave energy to lower frequencies.

If the aforementioned problematic data are excluded from the training and validation dataset (they are included in the results in Figures 1~4), the overall performance of the model will not be significantly improved (the overall RMSE only reduced by 0.02 m/s), because the number of samples for these corrupt data is very small compared to the overall sample size. However, the U10 RMSEs will be less than 1.5 m/s for all buoys at different locations. This indicates that the geographic dependence of the DNN model's error is weak. To further test the robustness of the DNN model in different locations, the training set and validation set were divided according to the buoys' locations. The data from buoys 45001-51101 (53 buoys) were selected as the training set and the buoys 41002-44066 (48 buoys) were selected as the validation set. The locations, wind-wave climate, and other environmental properties are significantly different for the two sets because none of the buoys in the validation set is in the same basin as the buoys in the training set. In this case, the established DNN model still has a

good performance in the validation set with an RMSE of ~1.15 m/s (the result can be seen in the reply to the reviewer in the online discussion).

    For wind directions (U10 > 7 m/s), the lowest RMSE is 7° and 68/94/100 out of the 101 buoys have RMSEs less than 14°/20°/22°, showing the robustness of the DNN model. The spatial distribution of RSME is similar to U10 RMSE (the CC between the RMSEs of U10 and wind directions is 0.51, significant at 99.9% level) with the lowest value in the open ocean at

low latitudes. The only buoy with RMSE larger than 22° is at Station 46082 (59.68°N,143.37°W). However, after a further check of the data, a bias of ~25° was found after 22-Sep-2018 (not shown), indicating there might be something wrong with the data themselves like the condition in Fig. 4. Similar conditions occur in some other buoys with RMSE > 20° (46001 and 44009). Two aforementioned buoys, 46087 and 46088, that are impacted by currents also have RMSEs > 20°. The reason for RMSE > 20° is unknown for the other two buoys, but errors of ~180° sometimes occur at the two buoys, largely increasing

the overall RMSE.

**4 Discussions**

    The wind information estimated from wave spectra achieves good accuracy, but the DNN model uses all available wave spectral information as the input. Usually, not all input terms are important for the model. Therefore, we tried to refine the DNN model using a sensitivity test. By blocking some of the inputs (setting the values of normalized input into zeros), one

can know which input is more important for the DNN model.

    Low-frequency waves are usually not coupled to the local wind, thus, the importance of different frequency bins was analyzed. The RMSEs after blocking some frequencies are shown in Fig. 5. For U10, it can be seen that inputs under 0.1 Hz are not important for the model, and blocking only one frequency bin has little impact on the result. However, blocking more bins at high frequencies, especially the bins near 0.2 Hz, has large impacts. For wind directions, it seems the inputs under 0.25

Hz are not important and the inputs near 0.38 Hz play the most important role in the model. Therefore, what the DNN learns from the data is a weighting average of the information from different frequencies. Voermans et al. (2020) also only considered the wave spectra higher than some frequencies in a spectrum, which is consistent with the model here.

    The importance of each of the Fourier coefficients was also analyzed. For the U10 (wind direction) DNN, the RMSEs after blocking $E$, $\alpha_1$, $\alpha_2$, $r_1$, and $r_2$ are 3.75, 1.17, 1.14, 1.47, and 1.20 m/s (17.3°, 111.9°, 16.2°,14.3°, and 14.4°, for U10 > 7

m/s), respectively. This indicates that $E$ and $\alpha_1$ are the most important parameters for estimating U10 and wind directions, respectively. This is in line with Voermans et al. (2020) where $E$ and $\alpha_1$ is the only parameter for the estimation of U10 and wind directions, respectively. Meanwhile, $r_1$ ($E$ and $\alpha_2$) seems to also play some roles in the estimation of U10 (wind directions). If we re-train the model with only $E$ ($\alpha_1$), the RMSE on the validation set can only reach 1.26 m/s (15.5°), slightly worse than the original model. This is probably because the $r_1$ contains the wave spreading information and the wave spreading at high

frequencies are also correlated to the wind speed, which can be used to slightly reduce the random error of the U10 from $E$ only. Similarly, $\alpha_2$ information can also partially reveal the wave direction in high frequencies, and $E$ is helpful to give the

energy weights for each frequency, which are helpful to reduce the random error of estimated wind directions. The above sensitivity test indicates that $E$ and $r_1$ above 0.1 Hz ($\alpha_1$, $\alpha_2$, and $E$ above 0.25 Hz) are the most important inputs for the estimation of U10 (wind directions), which is also in line with Voermans et al. (2020). Previous studies of wind remote sensing showed that the modulation of swells on *capillary waves* has some impacts on the wind speed retrievals (e.g., Stopa et al. 2016, Li et al. 2018, Jiang et al. 2020). Long swells also modulate short *wind-seas* (waves with relatively high frequencies measured by buoys, they are gravity waves instead of capillary waves). If this modulation process significantly impacts the buoy wind-estimation model, removing the long swell information will negatively impact the model accuracy. However, according to the results in Figure 5, the swell's modulation on *wind-seas* has little impact on wind estimation using buoy wave spectra. If we re-train a DNN using only these inputs (33×2=66 inputs for U10 and 17×3=51 inputs for wind directions) without changing other settings, the performance of the models is nearly the same as the original ones. The RSMEs stay less than 1.15 m/s and 14.5 °for U10 and wind directions, respectively, in 20 independent experiments.

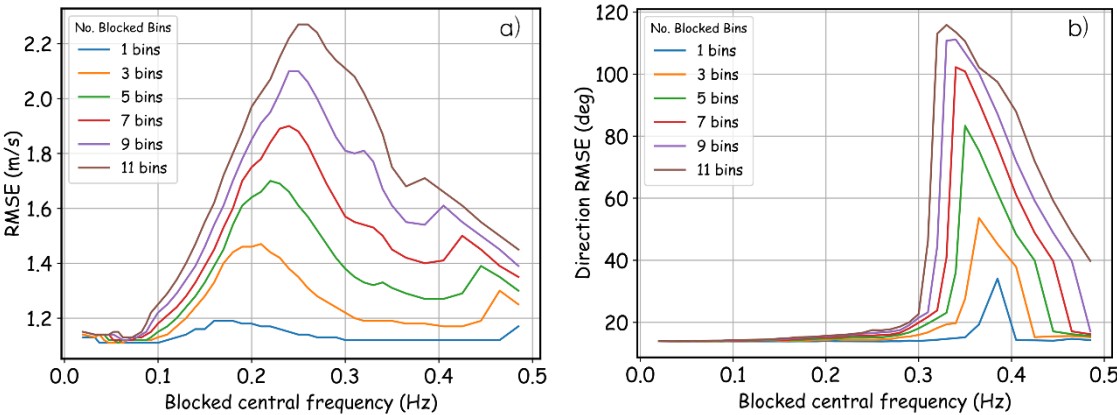

**Figure 5. (a) The RMSE between DNN-estimated and direct-measured U10 as a function of the blocked central frequency. Different colors indicate the results of blocking different numbers of bins. For example, the orange line indicates that the RMSE of the DNN model is ~1.45 m/s (the peak) when the input at 0.2 Hz and its two neighbouring bins, 0.19 and 0.21 Hz, are blocked (set to zero after normalization). (b) is the same as (a), but for the RMSE of wind direction.**

## 5 Concluding Remarks

Ocean wave spectra can be used to sea surface winds. Here, we trained two DNNs that can estimate U10 and wind directions ~40 minutes ago from high-frequency wave spectra. The overall accuracy of the wind-estimation DNN models is comparable with the state-of-the-art scatterometers under moderate U10. The two models can also be used as a quality control tool for wind and wave measurements from meteorological buoys.

The DNNs were trained using a large amount of data from only NDBC buoys but not compact wave buoys. However, applying the two models directly to compact wave buoy data (after interpolating the spectra from compact buoys into the frequency bins of NDBC buoys) will not result in significantly lower accuracy. This is because the DNN will automatically

select the NDBC wave spectra in the frequency with relatively high accuracy, and the accuracy of measured spectra from compact wave buoys is usually higher.

For the wave data from NDBC buoys, the performance of the U10 DNN is significantly biased when U10 is too high or too low, and the performance of the wind direction DNN becomes worse with the decrease of U10. Also, the accuracy of both models decreases when the surface currents are strong. We believe these shortcomings can be partly solved by compact wave drifters, resulting in better accuracy in estimating near-real-time wind properties. First, a smaller buoy size can resolve high-frequency wave spectra more accurately, which is helpful for wind estimation. Second, in the condition of strong wind or current, the moving velocity of the wave drifter is usually similar to that of the surface current, making the wavenumber and frequency spectra follow dispersion relation again in the buoy reference system. This can compensate for some of the errors induced by strong surface currents or wind-induced drifts. Therefore, significantly better accuracy can be achieved by training new DNN models with the spectral data (maybe also the drifting velocity data) from compact buoys using collocated wind and wave measurements. Such measurements can be obtained by placing some compact buoys near meteorological buoys or simply using the scatterometer or re-analysis wind as the training target.

Finally, we hope to point out that such DNN models need not to be trained from the beginning using a large amount of data. The DNN models presented in this paper can serve as pre-trained models which will significantly reduce the complexity of training the new models. With the compact wave buoys becoming increasingly widely used in observing wave parameters, their global network can be a new good-quality data source for both waves and wind after applying these models.

## Acknowledgments

This work is jointly supported by the Key Special Project for Introduced Talents Team of Southern Marine Science and Engineering Guangdong Laboratory (Guangzhou) (GML2019ZD0604), and the National Natural Science Foundation of China (U2006210, 41806010).

## Code/Data Availability

The NDBC data are available from the website of the National Centers for Environmental Information (https://www.ncei.noaa.gov/data/oceans/ndbc/cmanwx/). The two established wind-estimation DNN models are available as Python .plk files in the supplement materials where the corresponding example (as Python code) of implementing the two models are also available.

## Competing interests

The author declares no competing interests.

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
