# Peer review of "Wind Speed and Direction Estimation from Wave Spectra using Deep Learning"

_Atmospheric Measurement Techniques, 2021_

## Referee Comment (RC4)

Comments on AC1:

*Reply to Reviewer #1:*
* * *
*Indeed, as described in the preprint that the sea surface wind and waves are important parameters for the marine environment and ocean dynamics. This also implies that the interactions between them involve complex dynamic procedures resulting in the intricacies of coupling between them that make their individual characteristics difficult to resolve. Buoys on one hand though with limited amounts and distributions have been long providing good measurements of both wind and wave parameters respectively and simultaneously, on the other hand making complementary to remotely sensed wind and waves from satellites. The wind-wave interaction can then be modelled from buoy observations, while deep learning provides powerful tools in non-linear modelling and regression. The author thereby applies a deep neural network (DNN) for extracting wind information from wave spectrums provided by buoys for further applications to buoys without wind measuring ability benefitting such buoys with lower costs. The motivation and origin of this research are reasonable and good.*

First, the author would like to thank the reviewer for the positive opinion on the motivation of this work and the comments which are helpful for the improvement of the manuscript. Some revisions are made to the manuscript according to them. For some comments the author has different opinions on, explanations are given in this reply. The author hopes the reviewer can change his opinion and find the merit of this manuscript.

Of course, the interaction between wind and wave is very complicated, and the aim of this manuscript is not to reveal how waves (or to say, wind-sea) grow under the force of wind. The aim of the work is simply to establish a practical method of estimating wind speed and direction from wave spectrum measurements. The result of the model also indicates that this model to estimate sea surface wind from wave information can be useful. Therefore, the author submitted this work to *AMT* which is a journal about technology instead of a more "physical oceanography" journal such as JPO and JGR-Ocean.

Review comments to AC1 (ReAC1)_1: As stated in the previous comments, the validity of applications of AI in ocean science is challenging and triggers the recent highlighting of causality considered in the procedures of the model establishment. This is utterly lacking for this research, and not corrected from the replies. Besides, it is not likely that the result of this model described being useful. Moreover, AMT is about technology, correct applications of reasonable science are the fundamental of good technology, the arguments on science basis form the basis or the important part of the theories in applied technologies. Redo thoroughly this research is still necessary for many aspects. Specific comments are as following:

*Unfortunately, the research in this preprint falls into the trap set by that the DNN theory that can fit all models provided wide enough (though which is true mathematically). This may be due to ignoring that the meaning of the DNN model expressed is data and inputs-outputs dependent or self-consistent within such boundary. The model is only physically meaningful than mathematical results when not only the data or inputs are of good quality but also considering underlying physical principles to an extent a DNN can resolve. This can also be expressed with the state for one of the challenges for the application of artificial intelligence in ocean science: moving from purely statistical prediction to process-based models that embody causal relationships (Catalán, I., A. Solana, et al, 2021).*

The author does not think this work "falls into the trap" of DNN theory. Firstly, a model needs not to have explicit physical meaning to be useful. Nonparametric empirical models and methods are widely used in many aspects of ocean science. For example, the operational algorithms of many ocean remote sensors (e.g., the sea-state bias correction for altimetry, the D-Matrix algorithm for microwave radiometer, the watercolor algorithm for type II water, to name but a few) are also data-based and empirical. These models made many contributions to the development of ocean sciences and technologies. Secondly, the model presented in this work is not without physical bases or causal relationships. The work is based on the simplest idea that there is some quantitive causal relation between local wind and waves because waves are generated by wind. Although the explicit form of the wind-wave relationship is unknown, and the DNN can "learn" such a relationship using a large amount of data. In the author's opinion, artificial intelligence is the most suitable for such regression problems in ocean science: we know there are some causal relationships between inputs and outputs, but the physical model is too complicated to establish.

ReAC1_2: The problem of the proposed model lies in that the "some" causal relationships are without specific research but thrown into the DNN tools, which can result easily in applications in the unknown region the model cannot be representative. While the training sets cannot cover all different combinations of wind speed, wind direction, wave height/slope, wave direction, and the environmental parameter affecting different relations between them. Usually, in empirical models, such problems are seriously treated by specific analysis of the features and to how much extent the inputs and outputs are related, and what inputs cannot be applied. For example, empirical sea-state bias correction for altimetry is generally based on models of specific air-sea interaction as well as surface scattering methods of electromagnetic waves. As for 'D--Matrix' approach which seeks a linear relationship between measured SSM/I brightness temperatures and environment parameters, it is rather complex and uses matrix coefficients based n particular seasons and latitude bands that the measurements were taken from, and has a root in the measuring principles of radiometers. While there's no proper reason that the applications of the more powerful AI methods for regression excel requirements in this point. At the same time, the author argues about "a model needs not to have explicit physical meaning to be useful", and uses observed directly and unexplained in all aspects of spatiotemporal and statistical features of the inputs and outputs.

*More specifically for this research, it applies the spectrum parameters all at once as inputs for wind speed and direction ignoring the underlying multi-scale heterogeneity in time and space due to the complex relation of the interface interaction that can be embodied by a spectrum interpreting them in different approximations of the governing equation for energy distributed for different k values. Such approximation cannot only be expressed in another way round by expanding the observed energy distributed for different k values (spectrums are fitting of the observations) for another fitting from DNN. In other words, here DNN makes little extra contribution than the observed spectrum from this research. What is captured by DNN cannot be clarified makes things worse. Around Line 115, from the results, "the wave spectrum might also better reflect wind information a short period before" is contradictory to the fact that wind-wave spectrum ranges from lower frequencies to higher frequencies due to momentum transformation between waves of different lengths. For wind estimation, short wave measurement is relevant while the modulations from longer waves are non-negligible, from tilting effects to the short waves or modification of amplitudes of the short waves by exchange of energy altering atmosphere conditions close to the sea surface.*

The author does not understand why "DNN makes little extra contribution than the observed spectrum from this research". The DNN here is simply a model to estimate wind information from the input wave spectrum. This task might be done without the DNN, but no other models can perform as well as DNN at this stage, as far as the author knows (Please let the author know if there is a better model). Also, the author is confused about "spectrums are fitting of the observations". The observation of what? According to the understanding of the author, the wave spectra from the buoy can already be regarded as the observations of waves instead of a fitting.

The author believes that the expression, "the wave spectra might also better reflect wind information a short period before", is not contradictory to the fact that wind-wave spectrum ranges from lower frequencies to higher frequencies due to momentum transformation between waves of different lengths. But the author agrees that the expression is not precise and might cause some misunderstanding. Here, the "wave spectrum" only means the wind-sea spectrum measured by buoys, which needs some time to respond to the wind force (under the action of wind input, dissipation, quadruplet wave-wave interaction, etc.). Since the wind-sea spectrum is more impacted by the wind information a short period before than by the current wind, it is OK to say that the buoy wind-sea spectrum can better reflect the wind information a short period before than the current wind. The sentence has been revised to "the wave spectra of gravity waves from buoys might also better reflect the wind information a short period before than the current wind information", which should be more accurate. Our results showed that the DNN perform better when using the wave spectra to estimate the wind information one hour ago than to estimate the current wind, which also supports the above opinion. Of course, the "short period" here is not a specific time and can be from several minutes to several hours. Here we use the wave spectra to estimate the wind information one hour ago simply because the temporal resolution of the data is one hour.

In fact, the main reason for the author to directly apply the spectrum parameters all at once as inputs of the DNN is precisely to take into account the modulations of longer swells on shorter wind seas (another reason, which is not that important, is to illustrate how easily such a practical wind-estimation model can be established). If the modulation of swells is important for the estimation of wind information from buoy wave spectra, their impacts will be easily "learned" by the DNN (as demonstrated in many studies of wind remote sensing, e.g, Stopa et al. 2016 [Scatterometer], Li et al. 2018 [SAR], Jiang et al. 2020 [Altimeter]), and the low-frequency part will be important inputs of the wind-estimation model. However, the sensitivity test in Section 4 shows that the spectral information at frequencies lower than 0.1 Hz (mainly swells) does not have a significant impact on the model output, which indicates such modulations are not crucial for the estimation of wind from the wave spectrum. We have now pointed it out explicitly in the text that "Previous studies of wind remote sensing showed that the modulation of swells on capillary waves has some impacts on the wind speed retrievals (e.g., Stopa et al. 2016, Li et al., 2018, Jiang et al. 2020). However, according to the results here, the swell's modulation on wind-seas has little impact on wind-estimation using buoy wave spectra."

ReAC1_3: Again, the problem does not lie in if there is a more powerful model than DNN for complex problems (despite there being many other AI methods suitable for even chaotic situations), but the way in which this research is modeling. In addition to the previous comments, even if now it is clarified that the wave spectrum specifically refers to as the wind-sea partition the problem still exists. Here is a more detailed explanation as it seems a bit brief in the previous version.

For most of the NDBC buoys, the directly measured parameters are not the spectrum parameters. Note the ocean waves in different lengths embracing each other in a complicated way that is apparently non-linear and is still without a final answer due to unresolved air-sea interaction for wavelength ranging in a range of spatial scales. Hence the transfer from measurements of buoys to spectrum in different frequencies include basic assumptions on their interactions (the hourly wave height measurements from NDBC are not enough for an exact wave spectrum). And different empirical spectrums can be classified into this category. Although there is some new type of NDBC buoys that measure spectrums directly (the $\alpha$ parameter, et al.), and the amount is about 100 around half-half near-shore and off-shore. The samples are far from enough to cover all value space of different combinations of what also mentioned above as effecting factors for relating winds from waves: the interaction of air to sea, and the energy transfer as well as respond interactions between waves of different lengths: the near-surface air condition, wind speeds, wind directions, wave speeds, wave directions, et al., to form a steady model for such a complex problem facing only slightly better situations modeling wind and wind-induced waves in issues for calculating spectrum in the buoys when they cannot be measured. Besides, the off-shore and near-shore regions are with different features, and the locations are also limiting the conditions of sampling, in addition to the fatal lack of analysis of data inputs and outputs as well as related analysis (the comparison with the remotely sensed wind will be discussed in the next comment), how it can provide predictions from limited samples are not obtained from the established DNN model. Then this research is making no extra contribution than the spectrum coefficients from limited sampling. Though by applying parameters mimic to [1] helps narrow down the uncertainty space, while the lack of sampling can cause problems. And the conclusion that "the swell's modulation on wind-seas has little impact on wind estimation using buoy wave spectra" may also be due to the defect of the model established, while in [1], this is also considered in the parameter $\beta$.

[1] Voermans, J. J., Smit, P. B., Janssen, T. T., and Babanin, A. V.: Estimating Wind Speed and Direction Using Wave Spectra, Journal of Geophysical Research: Oceans, 125, 10.1029/2019jc015717, 2020.

*Moreover, though the training procedure is mathematically accomplishable, as in the preprint, where the results can be validated in error analysis from the testing set. Let alone the comparison of results to remotely sensed winds are not validated ignoring representative features of remote sensing results and buoy observations. Buoys generally provide the spot-based measurement of winds while remote sensing results are averages of a large region. The distributions of samples for each wind (and direction) bin are not discussed, the sample number may be skewed due to distributions of nature winds, while such effects are ignored in this research.*

The comparison with buoy-measured wind is almost a common practice in the validation of wind products from different types of remote sensors (e.g., scatterometer, altimeter, SAR). The community understands there is a representativeness error between remote sensing results and buoy observations (also between different remote sensors because they cannot measure exactly the same region at exactly the same time). The author even has a paper focus on mitigating the impact of this issue (Jiang 2020). However, buoy and remote sensing data are still comparable because of the potential equivalence between (remote sensing) spatial and (in-situ) temporal average, and also because most geophysical parameters do not vary severely in a small spatial-temporal domain. Otherwise, the comparison between any data from different types of data sources (in-situ, remote sensing, numerical model, etc.) will be problematic, which is not helpful for the development of science and technology.

The wind direction is relatively uniform as seen in Figure 2d/2f. The condition that wind speeds are not uniformly distributed can lead to the results that the model performs the best near the peak of the probability distribution. However, the empirical probability distribution function has been shown in Figure 2. The results indicate that the trained DNN model performs not that well for extreme winds (e.g., RMSE > 3 m/s for U10 > 20 m/s), due to insufficient numbers of samples in high wind speed. This is not surprising as the air-sea interaction becomes much more complicated during extreme wind (e.g., spray). The errors of estimated wind speed and direction as the function of measured wind speed have been shown in Figure 2c and 2f, which gives more details of the model's error for different wind speeds, which is also a guidance to the user of the model. We can see from Figure 2c that the model has the best performance for 3~10 m/s wind speed, and the RSME remains lower than 2 m/s for 1~17 m/s wind speed. This indicate the skewed sample numbers does not have large impacts on the model in moderate wind conditions.

ReAC1_4: The previous comment on this issue was brief and the point was not made clear, sorry about that! The comparisons of remotely sensed winds and buoy winds are typical and useful. However, in the research, the buoy wind is applied directly for matching with the scatterometer products, and reasons are as provided before, buoy winds are instant measurements while the remotely sensed winds are spatially averaged, both values cannot be compared directly, pre-processing are required. Meanwhile, this differs from the SSH measurements of buoys in that they can be compared with their nearest spatiotemporal remotely sensed match, for the SSH recorded are time-averaged values that are somehow equivalent with spatially averaged measurement. This is also the case for the wave period parameters.

*Although some sensitive analysis for inputs as the selection of frequency discussed in part 4, this was unfortunately misinterpreted as well, due to the little effort taken for understanding the relation between observed inputs and outputs. This is similar to the results part around line 115, longer waves are with wind information that cannot be resolved by the mapping to winds from the DNN established directly fitting the observations.*

As mentioned in the above response, the information of long waves is used as the inputs of DNN. If the reviewer is familiar with machine learning, he/she will know that the impact of swell on wind estimation can be easily captured by a DNN. The author is not saying that longer waves (frequency < 0.1 Hz) are without wind information, but the DNN results tell us the spectra of low-frequency waves provide no additional help for the estimation of wind speed and direction.

ReAC1_5: See ReAC1_3.

*The discussions following such content are not proper as well. When the model boundary is not clear due to the aspects listed above, there is little chance for these DNN models to apply in QC procedures or other applications. The results are also not likely to be improved including compact wave drifters, as the air-sea interaction in different scales is not likely to be well described in the reasons above.*

The author is not sure about what the reviewer means about the model boundary. But the author does not want to argue too much on whether this DNN model can be applied in QC procedures. The data in Figure 3 has already proved that some bad-quality data is identified using the DNN. This is the best evidence to show that the usefulness of this model in the QC of buoy data. It is noted that these bad-quality data were not identified in the QC procedures of the National Data Buoy Center.

The author needs to emphasize that the function of this DNN model is to estimate wind information from wave spectra instead of gives a better explanation about the physics of air-sea interaction. Regarding whether the results can be improved including compact wave drifters, the manuscript has shown that the high-frequency information is crucial for buoy-wave-spectrum-based wind estimation and the accuracy of the model is impacted by the ocean current. The data from such drifters can contain better-quality wave spectra (due to their better response to short waves) with more high-frequency information and also the ocean current information. For DNNs, better and more relevant inputs can usually give better output. That is why the author believes the results can be improved.

ReAC1_6: In addition to previous comments, the QC of a model not well established doesn't help.

*To wrap up, for such a model without awareness of the causalities between the inputs and outputs, especially under the circumstances such causalities are complex and wraps between even inputs and outputs, the deductions made based on them can easily go wrong. This is exactly the case for wind-wave interactions, such that improper analysis generally appears here and there for this preprint.*

As mentioned in a previous response, the causality between input and output is considered, but not in any explicit form. The author does not deny that this model can go wrong sometimes, especially in very low and extreme wind speeds. However, this has been discussed in the manuscript and the error functions of wind speed and direction were given. As a model to estimate wind speed and direction, there is no need to judge right or wrong, there is only accurate and inaccurate. As a famous saying goes, "all models are wrong, but some are useful". This model provides an accuracy of ~1.1 m/s for wind speed and ~14° for wave direction, which should be regarded as useful.

ReAC1_7: See previous comments. Besides, the saying is a warning to not stray into the mistake of choosing one's model as correct over reality. This is consistent with ReAC1_1 somehow.

*2) Around line 30, as mentioned before, short gravity-capillary waves are modulated by longer waves, though in the case of scatterometry, the orbital velocity of longer waves cannot be observed, and the tilting effect may not be obvious for them modulated to gather on the crests, by modulating the surface wind stress that changes the amplitudes of the short waves, which cannot be ignored, the long wave information does exist in scatterometer observations.*

Many studies have shown that the modulation of longer waves can impact the wind estimation of scatterometers. But it seems to be difficult to retrieve wave information directly using these impacts. To the best of the author's knowledge, there is no effective model that can obtain wave information from the scatterometer backscatter data independently. Therefore, it is OK to say wave information is not available from scatterometers.

ReAC1_8: In fact, they are available in scatterometer observations, for example:

a) Wright, J.: Backscattering from capillary waves with application to sea clutter, IEEE Transactions on Antennas and Propagation, 14, 749-754, 10.1109/tap.1966.1138799, 1966.
b) Plant, W. J.: in: Surface Waves and Fluxes, Springer, Dordrecht, https://doi.org/10.1007/978-94-009-0627-3_2, 1990.
c) Quilfen, Y., Chapron, B., Collard, F., and Vandemark, D.: Relationship between ERS Scatterometer Measurement and Integrated Wind and Wave Parameters, Journal of Atmospheric and Oceanic Technology, 21, 368-373, 10.1175/1520-0426(2004)021<0368:rbesma>2.0.co;2, 2004.

> *4) Around line 35, low temporal resolutions do not cause low performance near shore. There are near-shore products from scatterometers for example. Besides, inter-constellation will solve the coverage problems to an extent.*

> The author simply wants to state: 1) space-borne remote sensors often have limited temporal resolutions, 2) space-borne remote sensors often perform badly near shore. That is why an "and" is used instead of "so that". To make this point clearer and accurate, this sentence is revised to "…space-borne remote sensors often have limited temporal resolutions and they often perform worse in nearshore regions…".

> Of course, more satellites can increase the temporal resolution and spatial coverage. This is common sense that is not related to the theme of this manuscript so that the author thinks there is no need to mention it.

ReAC1_9: "1) space-borne remote sensors often have limited temporal resolutions," is also common knowledge should not mention if "more satellites can increase the temporal resolution and spatial coverage." is not needed.

2) Near shore are not necessarily poor or worse in performance, there are already examples many years ago:

a) Chelton, D. B., Schlax, M. G., Freilich, M. H. & Milliff, R. F. (2004). Satellite Measurements Reveal Persistent Small-Scale Features in Ocean Winds. Science, 303, 978--983. doi: 10.1126/science.1091901

b) Chelton, D. B., Freilich, M. H., Sienkiewicz, J. M., & Von Ahn, J. M. (2006). On the Use of QuikSCAT Scatterometer Measurements of Surface Winds for Marine Weather Prediction, Monthly Weather Review, 134(8), 2055-2071

> *5) Around line 45, again, direct comparisons of buoy results are remote sensing products are not validated.*

> This point has been explained in the previous response. The comparison between buoy and remote sensing results is a common practice. If this is not valid, the comparison of almost any data from two different sources will be invalid.

ReAC1_10: It is not validated for this research since the comparisons are not properly done, but not due to the theory to make comparisons between buoys and RS results are invalid. See also ReAC1_4.

---

## Author Comment (AC1)

*Reply to Reviewer #1:*
* * *
*Indeed, as described in the preprint that the sea surface wind and waves are important parameters for the marine environment and ocean dynamics. This also implies that the interactions between them involve complex dynamic procedures resulting in the intricacies of coupling between them that make their individual characteristics difficult to resolve. Buoys on one hand though with limited amounts and distributions have been long providing good measurements of both wind and wave parameters respectively and simultaneously, on the other hand making complementary to remotely sensed wind and waves from satellites. The wind-wave interaction can then be modelled from buoy observations, while deep learning provides powerful tools in non-linear modelling and regression. The author thereby applies a deep neural network (DNN) for extracting wind information from wave spectrums provided by buoys for further applications to buoys without wind measuring ability benefitting such buoys with lower costs. The motivation and origin of this research are reasonable and good.*

First, the author would like to thank the reviewer for the positive opinion on the motivation of this work and the comments which are helpful for the improvement of the manuscript. Some revisions are made to the manuscript according to them. For some comments the author has different opinions on, explanations are given in this reply. The author hopes the reviewer can change his opinion and find the merit of this manuscript.

Of course, the interaction between wind and wave is very complicated, and the aim of this manuscript is not to reveal how waves (or to say, wind-sea) grow under the force of wind. The aim of the work is simply to establish a practical method of estimating wind speed and direction from wave spectrum measurements. The result of the model also indicates that this model to estimate sea surface wind from wave information can be useful. Therefore, the author submitted this work to *AMT* which is a journal about technology instead of a more "physical oceanography" journal such as JPO and JGR-Ocean.
* * *
*Unfortunately, the research in this preprint falls into the trap set by that the DNN theory that can fit all models provided wide enough (though which is true mathematically). This may be due to ignoring that the meaning of the DNN model expressed is data and inputs-outputs dependent or self-consistent within such boundary. The model is only physically meaningful than mathematical results when not only the data or inputs are of good quality but also considering underlying physical principles to an extent a DNN can resolve. This can also be expressed with the state for one of the challenges for the application of artificial intelligence in ocean science: moving from purely statistical prediction to process-based models that embody causal relationships (Catalán, I., A. Solana, et al, 2021).*

The author does not think this work "falls into the trap" of DNN theory. Firstly, a model needs not to have explicit physical meaning to be useful. Nonparametric empirical models and methods are widely used in many aspects of ocean science. For example, the operational algorithms of many ocean remote sensors (e.g., the sea-state bias correction for altimetry, the D-Matrix algorithm for microwave radiometer, the watercolor algorithm for type II water, to name but a few) are also data-based and empirical. These models made many contributions to the development of ocean sciences and technologies. Secondly, the model presented in this work is not without physical bases or causal relationships. The work is based on the simplest idea that there is some quantitive causal relation between local wind and waves because waves are generated by wind. Although the explicit form of the wind-wave relationship is unknown, and the DNN can "learn" such a relationship using a large amount of data. In the author's opinion, artificial intelligence is the most suitable for such regression problems in ocean science: we know there are some causal relationships between inputs and outputs, but the physical model is too complicated to establish.

*More specifically for this research, it applies the spectrum parameters all at once as inputs for wind speed and direction ignoring the underlying multi-scale heterogeneity in time and space due to the complex relation of the interface interaction that can be embodied by a spectrum interpreting them in different approximations of the governing equation for energy distributed for different k values. Such approximation cannot only be expressed in another way round by expanding the observed energy distributed for different k values (spectrums are fitting of the observations) for another fitting from DNN. In other words, here DNN makes little extra contribution than the observed spectrum from this research. What is captured by DNN cannot be clarified makes things worse. Around Line 115, from the results, "the wave spectrum might also better reflect wind information a short period before" is contradictory to the fact that wind-wave spectrum ranges from lower frequencies to higher frequencies due to momentum transformation between waves of different lengths. For wind estimation, short wave measurement is relevant while the modulations from longer waves are non-negligible, from tilting effects to the short waves or modification of amplitudes of the short waves by exchange of energy altering atmosphere conditions close to the sea surface.*

The author does not understand why "DNN makes little extra contribution than the observed spectrum from this research". The DNN here is simply a model to estimate wind information from the input wave spectrum. This task might be done without the DNN, but no other models can perform as well as DNN at this stage, as far as the author knows (Please let the author know if there is a better model). Also, the author is confused about "spectrums are fitting of the observations". The observation of what? According to the understanding of the author, the wave spectra from the buoy can already be regarded as the observations of waves instead of a fitting.

The author believes that the expression, "the wave spectra might also better reflect wind information a short period before", is not contradictory to the fact that wind-wave

spectrum ranges from lower frequencies to higher frequencies due to momentum transformation between waves of different lengths. But the author agrees that the expression is not precise and might cause some misunderstanding. Here, the "wave spectrum" only means the wind-sea spectrum measured by buoys, which needs some time to respond to the wind force (under the action of wind input, dissipation, quadruplet wave-wave interaction, etc.). Since the wind-sea spectrum is more impacted by the wind information a short period before than by the current wind, it is OK to say that the buoy wind-sea spectrum can better reflect the wind information a short period before than the current wind. The sentence has been revised to "the wave spectra of gravity waves from buoys might also better reflect the wind information a short period before than the current wind information", which should be more accurate. Our results showed that the DNN perform better when using the wave spectra to estimate the wind information one hour ago than to estimate the current wind, which also supports the above opinion. Of course, the "short period" here is not a specific time and can be from several minutes to several hours. Here we use the wave spectra to estimate the wind information one hour ago simply because the temporal resolution of the data is one hour.

In fact, the main reason for the author to directly apply the spectrum parameters all at once as inputs of the DNN is precisely to take into account the modulations of longer swells on shorter wind seas (another reason, which is not that important, is to illustrate how easily such a practical wind-estimation model can be established). If the modulation of swells is important for the estimation of wind information from buoy wave spectra, their impacts will be easily "learned" by the DNN (as demonstrated in many studies of wind remote sensing, e.g, Stopa et al. 2016 [Scatterometer], Li et al. 2018 [SAR], Jiang et al. 2020 [Altimeter]), and the low-frequency part will be important inputs of the wind-estimation model. However, the sensitivity test in Section 4 shows that the spectral information at frequencies lower than 0.1 Hz (mainly swells) does not have a significant impact on the model output, which indicates such modulations are not crucial for the estimation of wind from the wave spectrum. We have now pointed it out explicitly in the text that "Previous studies of wind remote sensing showed that the modulation of swells on capillary waves has some impacts on the wind speed retrievals (e.g., Stopa et al. 2016, Li et al., 2018, Jiang et al. 2020). However, according to the results here, the swell's modulation on wind-seas has little impact on wind-estimation using buoy wave spectra."
* * *
*Moreover, though the training procedure is mathematically accomplishable, as in the preprint, where the results can be validated in error analysis from the testing set. Let alone the comparison of results to remotely sensed winds are not validated ignoring representative features of remote sensing results and buoy observations. Buoys generally provide the spot-based measurement of winds while remote sensing results are averages of a large region. The distributions of samples for each wind (and direction) bin are not discussed, the sample number may be skewed due to distributions of nature winds, while such effects are ignored in this research.*

The comparison with buoy-measured wind is almost a common practice in the validation of wind products from different types of remote sensors (e.g., scatterometer, altimeter, SAR). The community understands there is a representativeness error between remote sensing results and buoy observations (also between different remote sensors because they cannot measure exactly the same region at exactly the same time). The author even has a paper focus on mitigating the impact of this issue (Jiang 2020). However, buoy and remote sensing data are still comparable because of the potential equivalence between (remote sensing) spatial and (in-situ) temporal average, and also because most geophysical parameters do not vary severely in a small spatial-temporal domain. Otherwise, the comparison between any data from different types of data sources (in-situ, remote sensing, numerical model, etc.) will be problematic, which is not helpful for the development of science and technology.

The wind direction is relatively uniform as seen in Figure 2d/2f. The condition that wind speeds are not uniformly distributed can lead to the results that the model performs the best near the peak of the probability distribution. However, the empirical probability distribution function has been shown in Figure 2. The results indicate that the trained DNN model performs not that well for extreme winds (e.g., RMSE > 3 m/s for U10 > 20 m/s), due to insufficient numbers of samples in high wind speed. This is not surprising as the air-sea interaction becomes much more complicated during extreme wind (e.g., spray). The errors of estimated wind speed and direction as the function of measured wind speed have been shown in Figure 2c and 2f, which gives more details of the model's error for different wind speeds, which is also a guidance to the user of the model. We can see from Figure 2c that the model has the best performance for 3~10 m/s wind speed, and the RSME remains lower than 2 m/s for 1~17 m/s wind speed. This indicate the skewed sample numbers does not have large impacts on the model in moderate wind conditions.

*Although some sensitive analysis for inputs as the selection of frequency discussed in part 4, this was unfortunately misinterpreted as well, due to the little effort taken for understanding the relation between observed inputs and outputs. This is similar to the results part around line 115, longer waves are with wind information that cannot be resolved by the mapping to winds from the DNN established directly fitting the observations.*

As mentioned in the above response, the information of long waves is used as the inputs of DNN. If the reviewer is familiar with machine learning, he/she will know that the impact of swell on wind estimation can be easily captured by a DNN. The author is not saying that longer waves (frequency < 0.1 Hz) are without wind information, but the DNN results tell us the spectra of low-frequency waves provide no additional help for the estimation of wind speed and direction.

*The discussions following such content are not proper as well. When the model boundary is not clear due to the aspects listed above, there is little chance for these DNN models to apply in QC procedures or other applications. The results are also not likely to be improved including compact wave drifters, as the air-sea interaction in different scales is not likely to be well described in the reasons above.*

The author is not sure about what the reviewer means about the model boundary. But the author does not want to argue too much on whether this DNN model can be applied in QC procedures. The data in Figure 3 has already proved that some bad-quality data is identified using the DNN. This is the best evidence to show that the usefulness of this model in the QC of buoy data. It is noted that these bad-quality data were not identified in the QC procedures of the National Data Buoy Center.

The author needs to emphasize that the function of this DNN model is to estimate wind information from wave spectra instead of gives a better explanation about the physics of air-sea interaction. Regarding whether the results can be improved including compact wave drifters, the manuscript has shown that the high-frequency information is crucial for buoy-wave-spectrum-based wind estimation and the accuracy of the model is impacted by the ocean current. The data from such drifters can contain better-quality wave spectra (due to their better response to short waves) with more high-frequency information and also the ocean current information. For DNNs, better and more relevant inputs can usually give better output. That is why the author believes the results can be improved.
* * *
*To wrap up, for such a model without awareness of the causalities between the inputs and outputs, especially under the circumstances such causalities are complex and wraps between even inputs and outputs, the deductions made based on them can easily go wrong. This is exactly the case for wind-wave interactions, such that improper analysis generally appears here and there for this preprint.*

As mentioned in a previous response, the causality between input and output is considered, but not in any explicit form. The author does not deny that this model can go wrong sometimes, especially in very low and extreme wind speeds. However, this has been discussed in the manuscript and the error functions of wind speed and direction were given. As a model to estimate wind speed and direction, there is no need to judge right or wrong, there is only accurate and inaccurate. As a famous saying goes, "all models are wrong, but some are useful". This model provides an accuracy of ~1.1 m/s for wind speed and ~14 ° for wave direction, which should be regarded as useful.
* * *
*There are also other defects in descriptions:*

*1) around line 25, the description of the lack of meteorological buoys may be inherited from the reference (Voermans et al, 2020), while this is only partly true. There are such buoys available in India (NIOT, National Institute of Ocean Technology) and China*

*(NOTC, National Ocean Technology Center). There are also publications applications applying such buoys though NOTC is currently not openly accessible.*

The word "almost" has been changed to "mostly" so that the expression should be more accurate: "meteorological buoys are very sparsely distributed and are mostly only available along the coastlines of developed countries".
* * *
*2) Around line 30, as mentioned before, short gravity-capillary waves are modulated by longer waves, though in the case of scatterometry, the orbital velocity of longer waves cannot be observed, and the tilting effect may not be obvious for them modulated to gather on the crests, by modulating the surface wind stress that changes the amplitudes of the short waves, which cannot be ignored, the long wave information does exist in scatterometer observations.*

Many studies have shown that the modulation of longer waves can impact the wind estimation of scatterometers. But it seems to be difficult to retrieve wave information directly using these impacts. To the best of the author's knowledge, there is no effective model that can obtain wave information from the scatterometer backscatter data independently. Therefore, it is OK to say wave information is not available from scatterometers.
* * *
*3) Around line 30, the measuring in the nadir of altimeters does not result in low spatial resolution cross-track, maybe the coverage cross-track is low was meant to express.*

The sentence has been revised according to the suggestion of the reviewer.
* * *
*4) Around line 35, low temporal resolutions do not cause low performance near shore. There are near-shore products from scatterometers for example. Besides, inter-constellation will solve the coverage problems to an extent.*

The author simply wants to state: 1) space-borne remote sensors often have limited temporal resolutions, 2) space-borne remote sensors often perform badly near shore. That is why an "and" is used instead of "so that". To make this point clearer and accurate, this sentence is revised to "…space-borne remote sensors often have limited temporal resolutions and they often perform worse in nearshore regions…".

Of course, more satellites can increase the temporal resolution and spatial coverage. This is common sense that is not related to the theme of this manuscript so that the author thinks there is no need to mention it.
* * *
*5) Around line 45, again, direct comparisons of buoy results are remote sensing products are not validated.*

This point has been explained in the previous response. The comparison between buoy and remote sensing results is a common practice. If this is not valid, the comparison of almost any data from two different sources will be invalid.
* * *
*6) After all listed above, It is difficult to believe this research is included in the projects listed around line 255.*

*In all, I suggest a rejection of this manuscript for publishing.*

The author does not think the nonacademic questions should be discussed in the peer-reviewing. Still, the author would like to thank the reviewer for all the comments.

---

## Author Comment (AC3)

The author would like to thank the reviewer for the helpful comments. Some revisions are made to the manuscript according to them. For some comments the author has different opinions on, explanations are given in this reply. The author hopes the revised manuscript is acceptable for the reviewer.
* * *
*The author assume that the referenced power law adequately describes the impact of boundary-layer stability, whereas the authors of the power law point out that it applies only to near neutral conditions. Such an assumption will often be valid for strong winds (U10 > 15 ms$^{-1}$), however for wind speeds <7 ms$^{-1}$ the departures from neutral conditions are likely to be substantial. Furthermore, buoys measure winds relative to the fixed Earth where as stress, which drives waves, is dependent on surface relative currents. For lower wind speed cases the impact of currents could be substantial. While this is mentioned later in the manuscript, it would be wiser to address it earlier and perhaps in the quality control of the input data.*

The author understands that power law is only valid for the condition of neutral stability. The main reason for the author to use the simplest power-law profile is to make the inputs and the targets of the model consistent with Voermans et al. (2020) which is probably the first paper trying to establish a model to estimate wind information from wave spectra. With the same inputs and the targets, the performance of the two models can be directly compared. Besides, the author thinks that the physical meaning of wind speed indirectly estimated from the wave spectra (sea surface state) is probably closer to the equivalent neutral wind speeds derived from sea surface backscatter (e.g., space-borne scatterometers and altimeters). Therefore, the air-sea stability-dependent wind profile was not used here although it can give a more accurate extrapolation of the real 10-m wind speed. The DNN model in this study only perform well for 3-20 m/s, where the differences of extrapolated 10-m equivalent neutral wind between different methods (e.g., power law, log, LKB) are much smaller than the error of the wind estimation model itself (with respect to standard deviation). Therefore, different 10-m wind adjust method will also give similar results in this study.

Regarding the impacts of the ocean current, indeed, they are important for the cases of low wind speed. The impacts include not only the relative wind effect (the stress of wind is dependent on the relative movement between wind and currents), but also the "relative wave effect", that is, the phase velocity of high-frequency waves and the current velocity are at the same order of magnitude during strong currents so that the dispersion relation will be distorted by the current. However, the current data are not available from the buoy data so that data with strong current cannot be discarded during

the quality control. The impact of currents can only be regarded as the noise for wind estimation from buoy wave spectra in this case. Therefore, the author still feels it might be better to introduce this effect in the discussion of errors.
* * *
*What (if any) quality control was applied to the data? Frankly, a paper should not be submitted without this information. If any quality control was applied, why was it applied and why is it likely to be sufficient? If it was not applied, then why is it not needed?*

The data has been already quality controlled by National Data Buoy Center (NDBC) where the data is provided. The detailed information on NDBC data quality control can be seen from https://www.ndbc.noaa.gov/qc.shtml. In the description of the NDBC buoy data, the manuscript mentioned that "Many buoys from the National Data Buoy Center (NDBC) coastal-marine automated network can provide quality-controlled in-situ wave and wind measurements". Therefore, the author simply removed the data with bad-quality flags, and this has been clarified in the revised manuscript: "After removing the data records with bad-quality flags, more than 1.7 million records….were used in this study".
* * *
*The Fourier characteristics of waves are poorly described and need to be much more clearly explained.*

The author is a bit confused about the "Fourier characteristics of waves". Generally, the wave spectrum, which is the Fourier characteristics of ocean waves, is obtained by applying the Fast Fourier Transform to the time series of displacement, azimuth, pitch, and roll of buoys. However, this is almost common sense of the wind-wave community that needs not be explained in the manuscript. The author feels this is probably not what the reviewer is referred to. Therefore, it will be nice if the reviewer can explain a bit more on this comment.

The author tried to give a more detailed explanation on the five Fourier coefficients from the buoys in the revised manuscript, which now reads: "The buoy wave data includes five Fourier coefficients of waves for different frequencies in the range of 0.02-0.485 Hz (47 frequency bins) derived from the translational or pitch-roll information of buoys. The five Fourier coefficients are wave variance spectral densities ($E$) which describe the wave energy for each frequency, mean and principal wave directions for each frequency ($\alpha 1$ and $\alpha 2$), and first and second normalized polar coordinate of the Fourier coefficients ($r1$ and $r2$) which describe the directional spreading about the main direction the for each frequency. The five Fourier coefficients of different frequencies are the minimum requirement to reconstruct the directional

wave spectrum." After this revision, the physical meaning of the five Fourier coefficients should be clear for the reader.
* * *
*Does the lack of approximately uniform distribution over the parameter space impact the quality of the results, particularly for conditions that are poorly sampled? Normally there is a very large impact, with the results only applying to the conditions near the peak of the probability distribution.*

Indeed, the lack of uniform distribution of wind speed will lead to large errors for the conditions that are poorly sampled. This will also lead to that the model performs the best near the peak of the probability distribution. These are also parts of the reasons for the author to have Figure 2c in the manuscript. Both of the two effects can be seen from Figure 2c of the manuscript where the error metrics were given as a function of wind speed. The error is the smallest (~1 m/s) for the wind speed of 2-10 m/s where we have the largest sample size and becomes large for extreme wind speed where very few samples are available. However, this is inevitable since it is difficult to a have large sample size in extreme wind cases, and the air-sea interaction becomes much more complicated during extreme wind (e.g., strong spray and surface wind-driven drifts). With Figure 2c, one can know not only the error property of the model but also the impact of the skewed distribution of wind speed. One can conclude from Figure 2c that the lack of approximately uniform distribution does not have a large impact on the performance of the model, at least for moderate wind speed between 2-17 m/s. Some revision has been made to the discussion on the error of wind speed DNN, and the above explanation has been included (L136-145 of the revised version).
* * *
*Are the different Fourier components combined to produce a better result? I assume so, but the math suggests otherwise.*

Yes, the combination of different Fourier components can produce a better result than only using only one set of Fourier components. The author fails to understand why the math suggests otherwise. It will be nice if the reviewer can explain a bit more on this point.
* * *
*Can the one hour delay be better demonstrated with statistics and an appropriate graphic? It should be possible to show this result in a manner than much more clearly illustrates the width of the peak correlation or a time offset in the DNN.*

This is a very good suggestion, and this figure is added to the manuscript as Figure 3 (Also shown below). The author also change the used wind speed data to the 10-minute resolution "continuous wind" from the buoy, according to the suggestion of Reviewer #3. Figure 3 indicates that the best correlation between DNN-estimated and direct-measured wind is under the condition of a time offset of 40-60 minutes.

[Figure]

**Figure R1. Figure 3 in the manuscript: (a) The RMSE and CC of the DNN-estimated wind speed as a function of lag time between wave and wind measurements (waves' end sampling time minus winds' end sampling time). (b) The RMSE of DNN-estimated wind direction as a function of lag time between wave and wind measurements for wind speed higher than 7 m/s.**
* * *
*Errors in the results are attributed to strong currents, but these errors are far larger than expected due to currents (at least in Figure 3a, and unlikely in 3b). Buoys don't survive long in such strong currents.  Please consider alternative explanations or find evidence that the currents do exist.*

The reviewer seems to misunderstand the condition in original Figure 3a and 3b. Actually, they have nothing to do with the ocean currents. The author used them to explain why the RMSEs for the two buoys shown in the Figure are large (the two buoys are two of the buoys with the largest overall RMSEs), and to show that this wind-estimation DNN model can serve as an additional quality control/monitoring method for wind and wave sensors on meteorological buoys. For example, in original Figure 3a, after 26-Jan-2014, the difference between the measured and estimated wind speed suddenly becomes very large. It is noted that the measured wind speed remains lower than 5 m/s for more than 15 days. This is unrealistic for ocean winds, therefore, there must be something wrong with the measured wind speed. However, these data are not screened out in the NDBC quality control procedure. A similar condition happens in original Figure 3b, where stable bias between the measured and estimated wind speed was suddenly observed. Because the DNN model is unbiased and time-independent, such a systematic underestimation or overestimation of U10 for a long period has to be attributed to the problem of either wind or wave sensor. Because the buoy data has been quality controlled by NDBC, such conditions of bad-quality data were only identified in the two cases in Figure 3. Even for the buoy the strongest impacts by currents, the

error is not that large, as shown in the following example (Station 46087 at the Strait of Juan de Fuca where tidal currents are strong, and this is one of the buoys with the largest overall RMSE of DNN-estimated wind speed):

[Figure]

**Figure R2. Time-series comparison of direct-measured (blue) and DNN-estimate (orange) wind speed for Station 46087 from 16-Oct-2014 to 15-Jan-2015.**
* * *
*In summary, the methodology needs to be greatly improved. The accuracy assessment should not be presented as an overall single value for the dataset, but rather as a function of wind speed. The explanation for the cause of large errors is highly unlikely to be correct, although I appreciate the authors efforts to provide an explanation. The lack of check the quality of the input data, the physics of the adjustment to a 10m wind, and poor assessment of the quality of data should be addressed.*

The author believes that most of the points in this paragraph of the reviewer comment have been covered in the above responses except for the accuracy assessment. Of course, it is more reasonable to describe the error as a function of wind speed, and that is exactly what has been done in the manuscript. The error as a function of wind speed is shown in Figure 2c and presented in many places in the manuscript. However, many people in the community are also used to using a "typical" number to describe the error, maybe for simplicity. For instance, we often say that the error of wind speed, wind direction, and wave height for NDBC buoys are 1 m/s, 10°, and 0.2 m, respectively (e.g., https://www.ndbc.noaa.gov/rsa.shtml). The ocean remote sensing community also often says, for example, that the scatterometers have ~1 m/s error of wind speed (There are many papers, data product handbooks, and even textbooks, saying so). Therefore, the author thinks it is OK to present an overall single value of RMSE somewhere in the text.

Again, the author thanks the reviewer for his/her helpful comments.

---

## Author Comment (AC4)

*Reply to Reviewer #3:*
* * *
*This study is clearly presented and well written. The objective is to improve upon the recent work of Voerman et al. (2020) that attempted to invert near-surface wind speed and wind direction from ocean wave buoy datasets provided by the NDBC network of coastal and offshore buoys. That previous study provided a thorough review of wind-wave interaction as it pertains to buoy measurements and this inversion. The present study bypasses the geophysical basis and instead focuses on a sort of brute force neural network (DNN) approach to the wind estimation task using the NDBC data archive of five freq. dependent Fourier coefficients that are used to approximate the directional gravity wave spectrum from long to intermediate scale surface waves (both swell and wind sea). The study appears to use data from the entire buoy station network to develop separate wind speed and direction algorithms, provides detail on the network training and several relevant DNN adjustments during the training process, and then results that show some promising capability to provide wave-buoy derived wind estimate that agree better with the buoys' anemometer measurements. They also find that the winds derived in this manner appear to lag behind the actual surface winds in time by 30-60 minutes - and thus their final algorithm estimates not the wind at the present time, but actually the wind that occurred one hour before. They also find, as did the recent Voermans et al. study, that their best algorithms still have limitations at lower and higher wind speeds where the wave information does not unambiguously relate to the wind.*

The author would like to thank the reviewer for the patience for reading the entire paper carefully and the encouragement. The comments from the reviewer are very helpful for the improvement of the study. Some revisions are made to the manuscript according to them. For few comments the author has different opinions on, explanations are given in this reply. The author hopes the revised manuscript is acceptable for the reviewer.
* * *
*While this paper does show some potential for a neural network algorithm that takes the basic directional wave information provided by NDBC and outputs wind information, it does not appear to move things too far forward from the Voermans study they follow on from and the low and higher wind speed regime limitations that were highlighted in that study. What it does illustrate is that a DNN can improve on the semi-analytical approach used in the previous investigation.*

The author admits that this study does not move things forward from Veormans et al. (2020) with respect to the geophysical basis of the wave spectrum-based wind-estimation model. However, the final aim of establishing such a model, in the author's opinion, is to have the ability to estimate the wind information as accurately as possible. Since the underlying physics and the possibility of establishing such a model have been discussed by Veormans et al. (2020), this study focuses on the improvement of accuracy.

Because the relationship between inputs (spectrum) and outputs (wind) can be highly nonlinear and there might be some $2^{nd}$-order effect that is difficult to be considered in the semi-analytical model, the author simply used the DNN model to "learn" the input-output relationship to obtain better accuracy. The author believes that DNN is the best suitable for such problems: we have some understanding of the relationships between inputs and outputs, but the detailed physical model is too complicated to establish analytically. The results show that this selection is not bad, the accuracy of the estimated wind is improved significantly from Veormans et al. (2020) in conditions of moderate wind speed (the overall RMSE for 3-20 m/s wind speed is improved from ~2 m/s to ~1.2 m/s without time delay and ~1 m/s with a 40-minute time delay).

Regarding low and higher wind speed regimes, the author believes that this is the problem of almost all indirect wind-estimation models and one of the challenges of almost all wind measurement technologies. For low wind speeds, the response of surface waves is too weak while the impacts of other geophysical noises might be strong. For high wind speeds, the air-sea interaction is complicated while we do not have sufficient samples (there are less than 100 samples for U10>21 m/s) to build a robust model. Still, compared to Veormans et al. (2020), the DNN model also performs slightly better in high and low winds. For example, the RSME for 1 m/s, 2 m/s, 3 m/s, 15 m/s, 17 m/s, and 20 m/s buoy wind speeds were all improved significantly: 3 m/s→ 2 m/s, 2.5 m/s→1.2 m/s, 2 m/s→1 m/s, 2.5 m/s→1.5 m/s, 3 m/s→2 m/s, and 4 m/s→3 m/s, respectively.

*The finding that there they is an apparent delay between the wind speed and the wave-inferred wind speed is not physically inconsistent with Voermans et al. (2020) Figure 9g where the wind acceleration is related to model error residuals. However, there is an additional issue for the authors to consider first. The wave buoy measurements provided by NDBC have a center time that is 30 min past the top of the hour with data collected +-10 min of that time. The authors do not clearly provide detail on the NDBC wind products they are using, but if that product is the stdmet product then the center time for that 8 min. avg wind estimate is at minute 46 (measurements made from 42-50). Thus there is an inherent 15 min offset with the hourly wave data leading the wind. This factor may also color why the previous wind measurement is more highly correlated with the wave-inferred winds. Finally, the NDBC network does contain a large number of continuous wind measurement buoys where winds are measured every 10 minutes. Thus the authors have the opportunity to investigate the actual lagged correlation between DNN wave-derived winds and the anemometer data with 10 min resolution and perhaps at varying wind speeds.*

The data used by the author is the archived data from National Centers for Environmental Information (https://www.ncei.noaa.gov/data/oceans/ndbc/cmanwx/) in NetCDF form, and the actual acquisition time of wind, waves, and continuous wind are provided separately using different dimensions. However, the suggestion from the

reviewer, using continuous wind to investigate the lagged correlation, is very helpful. This work was conducted in the revised version of the manuscript, and the result is shown in the new Figure 3 (Also shown here as Figure R1). It is found that the overall best error metrics for wind speed and wind direction were found at 40-50 minutes and 40-60 minutes before the end of the waves' end sampling time.

[Figure]

**Figure R1. Figure 3 in the revised manuscript: (a) The RMSE and CC of the DNN-estimated wind speed as a function of lag time between wave and wind measurements (waves' end sampling time minus winds' end sampling time). (b) The RMSE of DNN-estimated wind direction as a function of lag time between wave and wind measurements for wind speed higher than 7 m/s.**

Following the suggestion of the reviewer, the author also investigate the lag correlation at different buoy wind speeds, and the results are shown in Figure R2. For different wind speeds, the best correlations (minimum RMSE) for wind speed were all found at the time offset of ~40 minutes (the lag for U10>12 m/s is not significant). Therefore, using a simple offset of 40 minutes should be sufficient for the model. Based on this result, the DNN models were retrained using 40 minutes delay.

[Figure]

**Figure R2. The RMSE of the DNN-estimated wind speed as a function of lag time between wave and wind measurements for different wind speeds.**

*Regardless, this issue points outs that using a series of DNN models to sort this out is an indirect and poorly-posed reverse engineering approach to infer the growth or dissipation rate of wind waves, as well as an illustration of the fundamental limitation in the use of surface waves to provide accurate wind measurements under a full range of wind forcing and sea states discussed in Voermans et al. (2020.*

Yes, the author thinks this might be the potential problem of all methods based on artificial intelligence: it is difficult for them to directly tell us new physics. And with such a large number of samples (million level), the DNN model has probably reached the limit of estimating wind using surface wave (at gravity range). If this model cannot solve the high/low wind problem, probably neither can other models, unless we have more samples of extreme wind or have a wider range of high-frequency tails (probably also not very helpful as frequency spectrum at the tails is very strongly impacted by surface current).
* * *
*A significant concern related to this time delay is the need to explain the potential implications of their DNN-derived estimates for users such as forecasters. The final DNN models are tuned to give wind speed and direction from the hour before. Thus I believe the first sentence of the Concluding Remarks should clarify this point. I believe the authors should consider a revisit of this product. Perhaps they should provide statistics and models for two wind options, the nearest time wind and the previous hour winds?*

The author has revised the second sentence of the Concluding Remarks to "…DNNs that can estimate U10 and wind directions ~40 minutes ago from high-frequency wave spectra…", which should be clear now. Regarding the two options, the statistics of the nearest time wind model are shown in Figure s2a and 2d. However, the author did not emphasize the "nearest time wind" option for three reasons: 1) The data of one hour's delay (now only 40 minutes' delay) can already be regarded as near real-time, which can be very useful for the operational application such as forecast. 2) In fact, the DNN model to estimate "nearest time wind" also has a better agreement with the wind 40 minutes ago. Therefore, even if the application is very sensitive, the 40-minute-delay wind can be directly approximated to the "nearest time wind" with a similar accuracy to an ad hoc model. There is no need to use two models.
* * *
*The model sensitivity tests in the discussion section are an ad hoc revisit of the more in-depth work of Voermans et al. (2020) and previous work (e.g. Jusko et al., J. Phys Ocean. 1995). But simply withholding part of the frequency spectrum from the inputs does not provide new results. It confirms, as the authors note (lines 205-210), what has already been shown in terms of the importance of the higher frequency portion of the spectrum closer to the wind sea peak frequency and the tail of the spectrum. The authors appear to perform this test in the same way for all wind speeds and conditions and perform the RMSE assessments similarly for all winds. This is a course sensitivity test. Perhaps something more creative could be done to investigate the potential to modify inputs with a goal to improve performance at low and high wind speeds?*

There are also two reasons to do the sensitivity test. One is simply to refine the input of the model. The DNN was established in a very brutal way of including all Fourier coefficients at all frequencies as the input. Using such a sensitivity test can let us know which of them do contribute to the wind estimation. This will help us to make the size of the DNN smaller so that can be more easily trained. This sensitivity test also tells us that including the $r_1$ information (which describes the directional spreading for each frequency) is helpful for the estimation of wind speed probably because the directional spreading of high-frequency waves also contains the information of wind speed. In fact, the author also tried to establish a DNN model for U10 estimation with only wave spreading information ($r_1$ and $r_2$), and the resulted overall RMSE can also reach 2.2 m/s, as shown in Figure R3. Therefore, such a simple sensitivity test can still provide some new information.

[Figure]

**Figure R3. Scatter plot of collocated DNN-estimated wind speed using only wave spreading information ($r_1$ and $r_2$) as input and direct-measured wind speed.**

The other aim is to check whether the modulation of low-frequency waves on high-frequency waves has a significant impact on the model. Previous studies have shown that the modulation of low-frequency waves on capillary waves can be a $2^{nd}$-order factor for wind remote sensing (e.g., Stopa et al. 2016, Li et al. 2018, Jiang et al. 2020). However, the results in Figure 5 (original Figure 4) show that this modulation effect is not important for wind estimation from surface gravity waves.

**Reference:**

Jiang, H., Zheng, H., and Mu, L.: Improving Altimeter Wind Speed Retrievals Using Ocean Wave Parameters. IEEE J. Sel. Top. Appl. Earth Obs. Remote Sens., 13, 1917–1924, doi:10.1109/JSTARS.2020.2993559, 2020.

Li, H., Mouch, A., and Stopa, J. E.: Impact of Sea State on Wind Retrieval from Sentinel-1 Wave Mode Data. IEEE J. Sel. Top. Appl. Earth Obs. Remote Sens.,12, 559-566, doi: 10.1109/JSTARS.2019.2893890, 2018.

Stopa, J. E., Mouche, A., Chapron, B., and Collard F.: Sea state impacts on wind speed retrievals from C-band radars. IEEE J. Sel. Top. Appl. Earth Obs. Remote Sens., 10, 2147–2155, doi:10.1109/JSTARS.2016.2609101, 2017.

*More detailed information on the specific wind and wave buoy products that they used in training, their data filtering and quality control, and references describing the approach that NDBC uses to extract the directional wave Fourier coefficients should be provided.*

The data used in this study is the data archived in National Centers for Environmental Information, so that the data has been quality controlled by NDBC. The author did not do too much quality control for the data except for removing the records with bad-quality flags. More detailed information on the data products was provided and the corresponding reference of measuring Fourier coefficients (Steele et al. 1998) was also provided in the revised manuscript.

**Reference:** Steele, K. E., Wang, D. W., Earle, M. D., Michelena, E. D., and Dagnall, R. J.: Buoy pitch and roll computed using three angular rate sensor. Coast. Eng., 35, 123-139, 1998.
* * *
*Given what is observed in terms of data quality in the section surrounding Figure 3, is there any concern that such corrupt data are present in the training and/or validation datasets? Moreover, as noted in the next paragraph, it would seem to be obvious that the algorithm training set should not include buoys where there is strong known wave/current interaction such as 46087 and 46088. This would be a highly unusual case of wind-wave-current interactions that would not be desired in a general-purpose wind algorithm that only uses the 5 Fourier coefficients and no surface current data as inputs.*

The corrupt data are present in the training and validation set. However, the number of samples for these corrupt data is very small which can neither impact the training of the model nor the validation of the model (with respect to error metrics). Therefore, the author did not re-train or re-validate the model for simplicity.

Similarly, regarding the cases with strong currents, because the current data is not available from the buoy, it is difficult to remove the cases with strong currents. Even for the buoys 46087 and 46088, the currents are not always strong. Including them in the training/validation dataset will also have almost no impact on the results (Figure R4). Meanwhile, when the users are using this model, it is also difficult for them to know whether there are strong currents at the location of wave measurement. Therefore, the current is considered as a source of environmental noise for this model. The model is not inapplicable in conditions of strong currents, but the accuracy will slightly decrease.

[Figure]

**Figure R4. The same as Figure 2b and 2c in the manuscript, but the data from buoy 46087 and 46088 is excluded from the training and validation dataset.**

*Similarly, was there any consideration given to differentiating between coastal, offshore, and/or differing wind-wave climate buoys in the model input training sets to improve performance, for example at low or high wind speeds.*

In fact, the author has not only tried to differentiate the coastal and offshore conditions, but also tried to use the buoys' distances to the nearest coast as an input term of the DNN model. However, this consideration did not improve the model so that the author did not mention it in the manuscript. However, on the bright side, this also indicates that the generalization ability of the DNN model is good and the users do not need to deal with several models for different conditions.

Different wind-wave climates of buoys were not considered in the model. But differing the location of the buoys has some implications of the wind-wave climate. The author even tried to establish a DNN model for each buoy, which did not improve the model, either. According to the suggestion of the reviewer, the author also tried to using the climatology monthly wave height and wind speed as the input of the DNN. However, there is still no improvement.

*The authors seem to be interested to develop a wind measurement system that competes with a satellite scatterometer or altimeter, but this project is inherently dealing with in an in situ platform. Is not the goal to develop an in situ system that has precision and accuracy metrics similar to those of the 10 min averaged wind anemometers used at sea?*

Although the data is obtained from an in situ platform, the rationale of this model is more similar to satellite scatterometer and altimeter that use the surface wave properties to indirectly retrieve the wind. The author thinks that indirectly estimated wind should be compared with the indirectly estimated wind. Meanwhile, both scatterometer and altimeter are regarded as successful remote sensors for wind speed retrieval, especially the scatterometer. As an indirect estimation model, being comparable with a scatterometer indicates that this model is already practical for many applications such as model assimilation. That is why the author mentioned remote sensing in the text several times.

Of course, it will be nice if the precision of the model can be similar to the anemometers. However, it is difficult for a model trained against the anemometer data (which is regarded as the "ground truth") to reach the same accuracy. Another problem is that if there is no better "ground truth", it seems also to be difficult to judge whether the accuracy of an indirect wind-estimation model is better or worse or similar than the anemometer data.

---

## Author Comment (AC5)

*Reply to Reviewer #1 :*
* * *
*ReAC1_1: As stated in the previous comments, the validity of applications of AI in ocean science is challenging and triggers the recent highlighting of causality considered in the procedures of the model establishment. This is utterly lacking for this research, and not corrected from the replies. Besides, it is not likely that the result of this model described being useful. Moreover, AMT is about technology, correct applications of reasonable science are the fundamental of good technology, the arguments on science basis form the basis or the important part of the theories in applied technologies. Redo thoroughly this research is still necessary for many aspects.*

The author would like to thank the reviewer again for his helpful discussion. The author still thinks that AI is best suited for the problem that we know there are some causal relationships between inputs and outputs, but the physical model is too sophisticated to establish. For a problem, if the full causality and the underlying physics are well understood and parameterized, there is no need to introduce AI anymore. Therefore, for such a deep learning-based empirical model, whether such a model is useful or not should be judged based on the performance of the model instead of its underlying physics. Also, the author needs to point out that this is not a model without physics concerns. The physics background of estimating wind information from wave spectra has been discussed in Voermans et al. (2020) so that this work focus on the wind-estimation model itself.

*Voermans J. J., Smit, P. B., Janssen, T., and Babanin, A. V.: Estimating wind speed and direction using wave spectra. J. Geophys. Res. Ocean, 125, 2019JC015717, doi:10.1029/2019JC015717, 2020.*
* * *
*ReAC1_2: The problem of the proposed model lies in that the "some" causal relationships are without specific research but thrown into the DNN tools, which can result easily in applications in the unknown region the model cannot be representative. While the training sets cannot cover all different combinations of wind speed, wind direction, wave height/slope, wave direction, and the environmental parameter affecting different relations between them. Usually, in empirical models, such problems are seriously treated by specific analysis of the features and to how much extent the inputs and outputs are related, and what inputs cannot be applied. For example, empirical sea-state bias correction for altimetry is generally based on models of specific air-sea interaction as well as surface scattering methods of electromagnetic waves. As for 'D--Matrix' approach which seeks a linear relationship between measured SSM/I brightness temperatures and environment parameters, it is rather complex and uses matrix coefficients based n particular seasons and latitude bands that the measurements were taken from, and has a root in the measuring principles of radiometers. While there's no proper reason that the applications of the more powerful AI methods for regression excel requirements in this point.*

As mentioned before, the physics background of estimating wind information from wave spectra has been discussed in Voermans et al. (2020). Their results and analysis have shown that the wave spectra can be used as the input of an empirical model for wind estimation. However, there might be more factors that might be difficult to take into full account in the semi-analytical method (e.g., the swell modulation effects mentioned by the reviewer, although this study shows it is not very important). That is why the author used a DNN model and also why the DNN can give a robust estimation of the wind information. If only some irrelevant parameters are used to train the DNN, the DNN will not give a good result. In other words, if there is no reasonable physics sustaining this model, this model cannot work that well.

Regarding the two examples, sea-state bias correction and D-Matrix, the author did not mean that they are not without physical bases. In these models, the physics background is known to some extent but not completely known. There are also some factors that might be difficult to take into full account in the analytical or semi-analytical models. Therefore, after the theoretical studies have shown what can be used as the inputs of a model, scientists established effective empirical models to bypass the complicity of some high-order processes. For instance, after the theoretical analysis shows that the sea-state bias can be linked to the wave height (Hs) and backscatter cross-section ($\sigma 0$), a data-based empirical Hs-$\sigma 0$ look-up table can be used to estimate the sea-state bias.

In fact, the logic of establishing this DNN model is very similar to that of establishing the D-Matrix algorithm. Regarding the causal relationship between input and output in D-Matrix, we know that the change of geophysical parameters, such as SST, water vapor, and wind, will impact the received radiance of different channels. However, the analytical form of the relationship between them is difficult to know. Similarly, for this problem, we know that the wind will impact the buoy-measured wave spectrum, but the analytical form of the relationship is also difficult to know. At that age, the training of a DNN is much more difficult to train and the concept of AI is not that popular, thus, scientists assume the relationship to be linear and use in-situ observations to train the linear D-Matrix. Similarly, this study uses the DNN and a large amount of data to find the relationship between inputs and outputs. We know today that it is also OK (and even better) to use a DNN to establish the relation between geophysical parameters and the brightness temperature of different channels of radiometers.

The author is also aware that the DNN can be over-fitted and can be inapplicable for the condition that is not covered in the training dataset. That is exactly the reason why the data used in this study were divided into the training set and validation set, each containing 50% of the data. The evaluation of the model's performance was only conducted in the independent validation set. According to the concern of the reviewer regarding whether the model is applicable for different locations, the author also tried to divide the training set and validation set according to the buoys' location. We use the data from buoys 45001-51101 (53 buoys) as the training set and the buoys 41002-44066 (48 buoys) as the validation set. The locations, wind-wave climate, and other environmental properties are significantly different for the two sets (none of the buoys in the validation set is in the same basin of the data of the training set). However, the

results remain quite good as shown in the figure below (Figure R1). This again shows that the resulting model can adapt the condition for different regions, showing the robustness of the model.

[Figure]

**Figure R1.** *The same as Figure 2b and 2c in the manuscript, but the training and validation sets are different. In this figure, the data from buoys 45001-51101 are used as the training set and the buoys 41002-44066 are used as the validation set, and the comparison is only conducted in the validation set.*

Regarding what should not be used as the model input, it is one of the advantages of the DNN model. If one input term is not important for the output, the DNN can "automatically" ignore the impact of it given the sample number is large. For instance, we can add more irrelevant inputs for the wind-estimation DNN model, which will not impact the result as the weight of these inputs will be set to zero during the training process of the DNN.
* * *
*ReAC1_2: At the same time, the author argues about "a model needs not to have explicit physical meaning to be useful", and uses observed directly and unexplained in all aspects of spatiotemporal and statistical features of the inputs and outputs.*

"A model needs not to have explicit physical meaning to be useful" is just the general attitude of the author. The author wants to express that even if a model is not explicitly physical meaningful, it can still be useful sometimes. For example, the AI models of pattern and speech recognition still seem to be far away from the explicit physical meaning. However, these models are already widely used in many aspects.
* * *
*ReAC1_3: Again, the problem does not lie in if there is a more powerful model than DNN for complex problems (despite there being many other AI methods suitable for even chaotic situations), but the way in which this research is modeling. In addition to the previous comments, even if now it is clarified that the wave spectrum specifically refers to as the wind-sea partition the problem still exists. Here is a more detailed explanation as it seems a bit brief in the previous version. For most of the NDBC buoys, the directly measured parameters are not the spectrum parameters. Note the ocean waves in different lengths embracing each other in a complicated way that is apparently non-linear and is still without a final answer due to unresolved air-sea interaction for wavelength ranging in a range of spatial scales. Hence the transfer from measurements of buoys to spectrum in different*

*frequencies include basic assumptions on their interactions (the hourly wave height measurements from NDBC are not enough for an exact wave spectrum). And different empirical spectrums can be classified into this category. Although there is some new type of NDBC buoys that measure spectrums directly (the α parameter, et al.), and the amount is about 100 around half-half near-shore and off-shore. The samples are far from enough to cover all value space of different combinations of what also mentioned above as effecting factors for relating winds from waves: the interaction of air to sea, and the energy transfer as well as respond interactions between waves of different lengths: the near-surface air condition, wind speeds, wind directions, wave speeds, wave directions, et al., to form a steady model for such a complex problem facing only slightly better situations modeling wind and wind-induced waves in issues for calculating spectrum in the buoys when they cannot be measured.*

The wave spectrum parameters from NDBC buoys used in this study are all derived from the translational or pitch-roll information from the accelerometers and inclinometers onboard buoys. A fast Fourier transform is applied to the sensors' time series (~20 minutes) to transform the data from the temporal domain into the frequency domain. Therefore, the buoys used in this study are all new types of buoys referred to by the reviewer, and all the spectra used in this study are directly measured instead of fitted. Of course, the measurement of the spectrum (using any method) is based on the assumption of the quasi-stationary random process and weak nonlinearity, which is why the sampling time for wave measurement can neither be too long or too short. It is noted that the concept of the wave spectrum (and the Fourier theory) itself is based on the assumption of linear superposition of waves with different scales.

Although the reviewer thinks that the samples of these buoys might not be enough to cover all value space of different combinations of effecting factors, the sample size of more than 1 million records is not a small one. Besides, many values in the effecting factor space without samples can still be obtained using the interpolation and extrapolation ability of the DNN. This can also be illustrated using the result in Figure R1. The samples from the training set should not be able to cover all combinations of factors in the validation set, as they are data from different basins. However, the model still performs well in the validation set.

*Steele, K. E., Wang, D. W., Earle, M. D., Michelena, E. D., and Dagnall, R. J.: Buoy pitch and roll computed using three angular rate sensor. Coast. Eng., 35, 123-139, 1998.*
* * *
*ReAC1_3: Besides, the off-shore and near-shore regions are with different features, and the locations are also limiting the conditions of sampling, in addition to the fatal lack of analysis of data inputs and outputs as well as related analysis (the comparison with the remotely sensed wind will be discussed in the next comment), how it can provide predictions from limited samples are not obtained from the established DNN model. Then this research is making no extra contribution than the spectrum coefficients from limited sampling. Though by applying parameters mimic to [1] helps narrow down the uncertainty*

*space, while the lack of sampling can cause problems. And the conclusion that "the swell's modulation on wind-seas has little impact on wind estimation using buoy wave spectra" may also be due to the defect of the model established, while in [1], this is also considered in the parameter β*

Although the off-shore and near-shore regions are with different features, these features do not necessarily impact the estimation of wind from wave spectra. Figure 1 in the manuscript has shown that the accuracy of wind in off-shore and near-shore regions are not significantly different. If whether a buoy is off-shore or near-shore is significant for the wind estimation, the performance of the DNN will be improved by including this factor. The author tried to include the distance to coast (and also tried the condition near/off-shore using the 50km offshore criterion) as the input of the DNN, and the model did not give a better output. Similarly, if the swell modulation is important for such a wind-estimation model, the model's residuals should be significantly correlated to the swell information and such correlation can be easily identified by a DNN and included in the model.
* * *
*ReAC1_4: The previous comment on this issue was brief and the point was not made clear, sorry about that! The comparisons of remotely sensed winds and buoy winds are typical and useful. However, in the research, the buoy wind is applied directly for matching with the scatterometer products, and reasons are as provided before, buoy winds are instant measurements while the remotely sensed winds are spatially averaged, both values cannot be compared directly, pre-processing are required. Meanwhile, this differs from the SSH measurements of buoys in that they can be compared with their nearest spatiotemporal remotely sensed match, for the SSH recorded are time averaged values that are somehow equivalent with spatially averaged measurement. This is also the case for the wave period parameters.*

First, this research did not match the buoy wind with the scatterometer products. The collocation is only made between buoy wind and buoy spectra. Second, the wind measured by buoys are also time average so that can be compared with remote sensing (spatially averaged) wind product. As mentioned in the previous reply, such comparison is common practice in the validation of wind products of remote sensors including wind products from scatterometers and altimeters.
* * *
*ReAC1_6: In addition to previous comments, the QC of a model not well established doesn't help.*

Figure 3 in the manuscript (Figure 4 in the revised manuscript) has already proved that some bad-quality data is identified. These data were not identified in the NDBC QC procedure.
* * *
*ReAC1_7: See previous comments. Besides, the saying is a warning to not stray into the mistake of choosing one's model as correct over reality. This is consistent with ReAC1_1 somehow.*

This has been also mentioned in the reply to ReAC1_1. The physics background of estimating wind information from wave spectra has been discussed in Voermans et al. (2020) so that this work focus the wind-estimation model itself.
* * *
*ReAC1_8: In fact, they are available in scatterometer observations, for example:*
*a) Wright, J.: Backscattering from capillary waves with application to sea clutter, IEEE Transactions on Antennas and Propagation,14,749-754,10.1109/tap.1966.1138799, 1966.*
*b) Plant, W. J.: in: Surface Waves and Fluxes, Springer, Dordrecht, https://doi.org/10.1007/978-94-009-0627-3_2, 1990.*
*c) Quilfen, Y., Chapron, B., Collard, F., and Vandemark, D.: Relationship between ERS Scatterometer Measurement and Integrated Wind and Wave Parameters, Journal of Atmospheric and Oceanic Technology, 21, 368-373, 10.1175/1520-0426(2004)0212.0.co;2, 2004.*

The "wave information" in the context refers to information of gravity wave (instead of capillary wave) such as wave spectrum, wave height, wave period, or wave direction. These wave parameters are not available from the scatterometer data product. In the three reference provided by the reviewer, the wave parameters are also unavailable from scatterometers.
* * *
*ReAC1_9: "1) space-borne remote sensors often have limited temporal resolutions," is also common knowledge should not mention if "more satellites can increase the temporal resolution and spatial coverage." is not needed.*
*2) Near shore are not necessarily poor or worse in performance, there are already examples many years ago:*
*a) Chelton, D. B., Schlax, M. G., Freilich, M. H. & Milliff, R. F. (2004). Satellite Measurements Reveal Persistent Small-Scale Features in Ocean Winds. Science, 303, 978--983. doi: 10.1126/science.1091901*
*b) Chelton, D. B., Freilich, M. H., Sienkiewicz, J. M., & Von Ahn, J. M. (2006). On the Use of QuikSCAT Scatterometer Measurements of Surface Winds for Marine Weather Prediction, Monthly Weather Review, 134(8), 2055-2071*

According to the suggestion of the reviewer, this sentence has been revised to "space-borne remote sensors often perform worse in nearshore regions than in the open ocean due to the land contamination of backscatter" (the expression about the limited temporal resolution is removed from this sentence). Yes, the performance of remote sensors are not necessarily worse near shore, but the retrievals are often (not always) impacted by land contamination of backscatter near shore.
* * *
*ReAC1_10: It is not validated for this research since the comparisons are not properly done, but not due to the theory to make comparisons between buoys and RS results are invalid. See also ReAC1_4.*

This has been discussed in the reply to ReAC1_4. This study itself did not involve remote sensing data at all. The RMSEs between remote sensing and in-situ wind (~1 m/s and 15 °) are from many previous papers and is only used as an error reference for the model in this study.

The author thanks the reviewer again for these comments.

---

## Author Response (AR2)

**Response Letter**

*Ref.:    AMT-2021-279                Atmospheric Measurement Techniques*

*Title:    Wind Speed and Direction Estimation from Wave Spectra using Deep Learning*

*Author: Haoyu Jiang*

Dear Prof. Ad Stoffelen,

Thank you again for concerning the manuscript in *Atmospheric Measurement Techniques*. I appreciate you and the reviewers for your earnest work. The comments from the reviewers are very helpful, and the paper has been revised carefully according to some of them. The point-by-point response to the reviewers' comments are attached, as well as a tracked-changes version of the manuscript. For some comments I do not completely agree on, the explanation is also given in the point-by-point response.

I hope that this version of the manuscript is acceptable for publication. If you have any questions, please feel free to contact me. I appreciate your support very much.

Best regards,

Yours sincerely,

Haoyu Jiang

*Though the author replied the suggestions, the main issues are not resolved in the following aspects, which makes the manuscript not suitable for publication:*

The author would like to thank the reviewer again for his helpful discussion. However, the author does not completely agree with the comments from the reviewer. The explanation has been given in the following point-by-point response.
* * *
*1) The samples (buoys) are not enough in space distributions to guarantee the robustness of the relationship established by the DNN.*

The space distribution of the buoys is not always a key factor for the robustness of an empirical model. Otherwise, all remote sensing empirical algorithms or calibration relationships established using these NDBC buoys will lack robustness. Figure 1 in the manuscript has shown that the model is applicable for almost all buoy locations except for the region with strong tidal current (Station 46087 and 46088 at the Strait of Juan de Fuca, and even for them, the RMSE of U10 is still less than 2 m/s). If the spatial distribution of the buoys is insufficient to guarantee the robustness of the relationship established by the DNN, there will be some locations with much larger errors.

To further test the robustness of the DNN model in different locations, the author also tried to divide the training set and validation set according to the buoys' location. The data from buoys 45001-51101 (53 buoys) were selected as the training set and the buoys 41002-44066 (48 buoys) were selected as the validation set. The locations, wind-wave climate, and other environmental properties are significantly different for the two sets, because none of the buoys in the validation set is in the same basin as the buoys in the training set. However, the results remain quite good as shown in the figure below (Figure R1). This again shows that the resulting model can adapt the condition for different regions, which confirms that the robustness of the DNN model can be guaranteed.

Some of the above explanation has been added to the revised manuscript to make this point clearer for readers.

[Figure]

*Figure R1. The same as Figure 2b and 2c in the manuscript, but the training and validation sets are different. In this figure, the data from buoys 45001-51101 are used as the training set and the buoys 41002-44066 are used as the validation set, and the comparison is only conducted in the validation set.*
* * *
*2) The time delay between wind and buoy measured wave parameters is specifically discussed enough from neither the DNN model nor the physical principles.*

Although it is assumed that wind input, breaking dissipation, and wave-wave interaction are in equilibrium in the high-frequency part of the wave spectrum in Phillips (1985), a time delay should be within expectance because the high-frequency portion of the wave spectrum integrates preceding wind conditions. Therefore, it is natural to consider whether the buoy wave spectrum can better represent the local wind information some time ago. Using the data, the author proved that it is best to use the wave spectra to estimate wind speeds and directions ~40 minutes ago. The author understands that the "best" time lags can be different for different parts of the wave spectrum (Jiang and Mu 2019), and different past wind can correspond to different "best" time lags. However, the aim here is only to find a time lag to best link the local wind and waves from a statistical point of view, to make the algorithm be able to form a time series with a fixed time interval between estimations.

**Reference:**

Jiang, H., and Mu, L. (2019). Wave Climate from Spectra and Its Connections with Local and Remote Wind Climate. *Jouranl of Physical Oceanography*, 49, 543-559.
* * *
*3) The analogy with scattermeter data against buoys is not reasonable.*

The author needs to re-emphasize that this research did not match the buoy wind with the scatterometer products. This work only compared the wave-estimated wind and the

anemometer-measured wind. However, as the author mentioned before, wind measured by buoys is time average that is comparable with remote sensing (spatial average) wind product based on the equivalence of space and time variability. Such comparison is common practice in the validation of wind products of remote sensors, including those from scatterometers and altimeters. Many related publications are doing so.

*4) Lack of application prospects.*

Compact wave buoys are increasingly widely used in global wave observations. For example, more than 2,000 Spotter buoys have been deployed in global oceans by Sofar Ocean Technologies (locations of them can be viewed at http://weather.sofarocean.com/). Due to the compact size of such buoys, they cannot equip anemometers so that wind information is not directly available from them. The model can estimate the wind speed and direction with an accuracy of ~1.1 m/s from only wave spectrum information. Therefore, with compact wave buoys becoming increasingly widely used in observing wave parameters, they can also become a new good-quality data source of sea surface wind after applying the model presented in this study. Besides, the manuscript also showed that this model can also be used in the quality control of the wind and wave data from coastal meteorological buoys. Therefore, it is unreasonable to state that this model lacks application prospects.

**Responses to Reviewer2:**

*The authors have addressed the reviewer comments with sufficient attention and improved the manuscript. It is understood that a neural network approach to mapping between the buoy-derived spectral coefficients and the buoy winds is intended to provide a more optimized wind inversion, and not to expressly, or readily infer the geophysical basis for the mapping.*

*The results demonstrate a significant improvement to that of Voermans et al. (2020). This is not unexpected given the approach that uses all freq. band data and Fourier components for 101 buoys over four year, and given the likely large size of the resulting DNN. As the authors note in review discussions, this likely indicates the limits for how well the NDBC discus buoy data can estimate wind speed and direction.*

The author would like to thank the reviewer again for his/her encouragement and valuable comments. These comments are very helpful for the improvement of the manuscript, and revisions have been made according to them. A revised version of the manuscript with changes highlighted is also attached. The author hopes that the revised manuscript is acceptable for the reviewer.

Indeed, it is difficult to give the underlying physics of a DNN model especially when its input is a high-dimensional vector. Fortunately, the physics background of estimating wind information from wave spectra has been discussed in Voermans et al. (2020) so that this work can focus on the wind-estimation model itself. In the author's opinion, the DNN (or other AI methods) is the best suited for the problem that we know there are some causal relationships between inputs and outputs, but the accurate analytical physical model is too sophisticated to establish. That is why the author thinks that this wind-inversion problem is a good (this seems not to be humble, but the author does think so) application of the DNN and this work deserves to be published in a relatively technical journal.
* * *
*A remaining issue with the revised paper I see is the following...*
*The text near line 235-238 in the Discussion section should more clearly explain and support their conclusion that long wave modulation of capillary waves has little impact on wind estimation using buoy spectra. Their citations refer to wind remote sensing. But*

*this is a buoy wave vs. anemometer wind paper. Not a remote sensing paper.*

*Their assertion is off the mark. First, the NDBC wave buoys only accurately measure gravity waves up to f=0.4 Hz at best. These waves are much longer than those used in remote sensing and the buoys certainly do not measure capillary waves. Thus this suppostion that the DNN sensitivity tests illustrate that long wave-short wave interaction impacts on the model are absent is simply wrong. Rather - their sensitivity tests seem to imply that there is limited additional wind-correlated information carried in frequencies lower than 0.1-25 Hz when considering the overall training and validation datasets and this DNN approach. This is as much as one can conclude.*

This seems to be an expression problem and the reviewer seems to misunderstand the author's conclusion (it is the fault of the author, sorry for that). The author did not mean that long wave modulation of *capillary waves* has little impact on wind estimation using buoy spectra (the buoy does not provide capillary wave information so that this cannot be concluded). What the author wants to express is that the long wave modulation of *wind-sea* (i.e., waves with frequencies within 0.1-0.5 Hz which are all gravity waves instead of capillary waves) has little impact on wind estimation using buoy spectra. By the way, the author also did not say that the long wave does not modulate wind-seas. It is just that this process has little impact on wind estimation.

The rationale of this method has some similarities to wind remote sensing: both of them are using wave information to indirectly retrieve the wind information. The difference is that remote sensing uses capillary waves while buoys use gravity waves to estimate the wind. So that some processes can be linked in the two methods. The impact long wave modulation of *capillary waves* has been proved to be significant for remote sensing. Then, one will wonder whether the impact long wave modulation of high-frequency gravity waves measured by buoys is significant for wave-spectrum-based wind estimation. The answer from the author's result is a NO. Because this is probably the second work (the first is Voermans et al. (2020)) of using gravity waves to retrieve wind speed, it is a bit difficult to find other related references.

This part has been revised to make the expression more clear to avoid the potential misunderstanding: "Previous studies of wind remote sensing showed that the modulation of swells on *capillary waves* [here we use italic to stress the difference] has some impacts on

the wind speed retrievals (e.g., Stopa et al. 2016, Li et al. 2018, Jiang et al. 2020). Long swells also modulate short wind-seas (waves with relatively high frequencies measured by buoys, they are gravity waves instead of capillary waves). If this modulation process significantly impacts the buoy wind-estimation model, removing the long swell information will negatively impact the model accuracy. However, according to the results in Figure 5, the swell's modulation on wind-seas has little impact on wind estimation using buoy wave spectra."
* * *
*Minor comments:*
*I don't believe the authors have yet clarified inside the new draft that the problematic Jaun de Fuca buoy data is included in the model training and that they they have confirmed that these (and other problematic coastal buoys in their set) do not significantly or adversely affect model performance.*

This has been clarified in the new version of the manuscript (at Line 203): "If the aforementioned problematic data are excluded from the training and validation dataset (they are included in the results in Figures 1~4), the overall performance of the model will not be significantly improved (the overall RMSE only reduced by 0.02 m/s), because the number of samples for these corrupt data is very small compared to the overall sample size. However, the U10 RMSEs will be less than 1.5 m/s for all buoys at different locations."
* * *
*Near lines 123-125 there should be some relevant supporting citation and more precise quantification of the expected growth rate of the 0.2-0.4 Hz part of the wave spectrum and relevant time scales. "short period to grow" is imprecise. Gravity-capillary waves are not instant equilibrium with the wind. And certainly the wave buoy measurements contain information on the wind before. This is not a case of "might". Please provide appropriate citation or evidence.*

The growth of the high-frequency wave spectrum is mainly controlled by wind input, breaking dissipation, and wave-wave interaction, and this process responds rapidly to the local wind. In the open ocean, it is often assumed that $dE/dt = 0$ in the equilibrium range ($E$ is the wave spectrum, $t$ is the time) (Phillips 1985, which gives the shape of the spectrum tail as a function of U10). However, the author failed to find the supporting citation for

evaluating the relevant time scales of wave growth. That is why the author used such an imprecise way of describing this. Another problem here is that the sensitivity test in Section 4 has not been conducted yet in this part of the text, thus, the readers have not known which part of the wave spectrum is important for the estimation of wind information yet. That is why the author did not try to discuss the growth rate of the wave spectrum here.

As pointed out by the reviewer, the expression, "gravity waves (wind-sea) need a short period to grow", is imprecise (and even wrong). Considering the suggestion of the reviewer, the author changed the expression in this part to: "Different from the capillary waves with very high frequencies always in instant equilibrium with the local wind, the growth of gravity waves is time-dependent. Besides the current wind information, the wave spectrum measured by a buoy at a given location and time also contains remote and past wind information (Jiang and Mu 2019), because the wave spectrum is, to some degree, integrated winds. Therefore, *it is possible that the buoy wave spectrum can better represent the local wind information some time ago*. Based on this idea, the wave spectra were also collocated with past wind measurements using different time lags." Although the "some time ago" in the italic sentence is still imprecise, this sentence does not contain the physics of wave growth and should be a reasonable expression.

**Reference:**

Phillips, O. M. (1985). Spectral and statistical properties of the equilibrium range in wind-generated gravity waves. *Journal of Fluid Mechanics*, 156, 505–531.

Jiang, H., and Mu, L. (2019). Wave Climate from Spectra and Its Connections with Local and Remote Wind Climate. *Jouranl of Physical Oceanography*, 49, 543-559.
* * *
*Editorially, it seems peculiar to single out the two Juan de Fuca buoy datasets and ocean currents as a key issue to discuss or address in the concluding remarks, lines 251-258. Perhaps the overall issue of translating these DNN model results from the large NDBC buoys and their data to the emerging drifter wave buoy program should be introduced first. This is the third paragraph of the conclusion. Maybe switch these paragraphs and revise.*

According to the reviewer's suggestion, the author slightly changed the logic of writing the concluding remarks. Now the concluding remarks are divided into four paragraphs.

The first paragraph is a short summary: "Ocean wave spectra can be used to sea surface winds. Here, we trained two DNNs that can estimate U10 and wind directions ~40 minutes ago from high-frequency wave spectra. The overall accuracy of the wind-estimation DNN models is comparable with the state-of-the-art scatterometers under moderate wind speeds. The two models can also be used as a quality control tool for wind and wave measurements from meteorological buoys."

The second paragraph is the overall issue of translating these DNN model results from the large NDBC buoys and their data to the emerging drifter wave buoy program, as suggested by the reviewer: "The DNNs were trained using a large amount of data from only NDBC buoys but not compact wave buoys. However, applying the two models directly to compact wave buoy data (after interpolating the spectra from compact buoys into the frequency bins of NDBC buoys) will not result in significantly lower accuracy. This is because the DNN will automatically select the NDBC wave spectra in the frequency with relatively high accuracy, and the accuracy of measured spectra from compact wave buoys is usually higher."

The third paragraph is to briefly review the problems of the models using NDBC buoys and to predict that these problems can be partly solved by compact wave drifters: "For the wave data from NDBC buoys, the performance of the U10 DNN is significantly biased when U10 is too high or too low, and the performance of the wind direction DNN becomes worse with the decrease of U10. Also, the accuracy of both models decreases when the surface currents are strong. We believe these shortcomings can be partly solved by compact wave drifters, resulting in better accuracy in estimating near-real-time wind properties. First, a smaller buoy size can resolve high-frequency wave spectra more accurately, which is helpful for wind estimation. Second, in the condition of strong wind or current, the moving velocity of the wave drifter is usually similar to that of the surface current, making the wavenumber and frequency spectra follow dispersion relation again in the buoy reference system. This can compensate for some of the errors induced by strong surface currents or wind-induced drifts. Therefore, significantly better accuracy can be achieved by training new DNN models with the spectral data (maybe also the drifting velocity data) from compact buoys using collocated wind and wave measurements. Such measurements can be obtained by placing some compact buoys near meteorological buoys or simply using the scatterometer or re-analysis wind as the training target."

The last paragraph is the discussion about the prospect, which is still the last part of the original manuscript: "Finally, we hope to point out that such DNN models need not to be

trained from the beginning using a large amount of data. The DNN models presented in this paper can serve as pre-trained models which will significantly reduce the complexity of training the new models. With the compact wave buoys becoming increasingly widely used in observing wave parameters, their global network can be a new good-quality data source for both waves and wind after applying these models."
* * *
*Given that the concluding remarks suggest that their models could be built upon for future work, it would be useful to provide some access to the code and training sets that were used to develop them, or to provide some a subroutine or lookup table to implement their models. Is this included with the paper?*

This is a very good suggestion. Regarding the data of training and validation sets, all the buoy data used in this study are available from the National Centers for Environmental Information (https://www.ncei.noaa.gov/data/oceans/ndbc/cmanwx/). This has been clarified in the acknowledgement section. As the total size of the five years' buoy wind and wave data from ~100 buoys is about two GB, there seems to be no need to upload them again. Meanwhile, according to the suggestion of the reviewer, the established DNN models were uploaded together with the revised version of the manuscript as supplement materials, and the Python code (with the annotation) of using the two DNNs was also attached. With these files, the readers/users can easily implement the models.

We also added the following statement to the acknowledge part: "The two established wind-estimation DNN models are available as Python .plk files in the supplement materials where the corresponding example (as Python code) of implementing the two models are also available."
* * *
*There are numerous grammatical issues with the revision. These are minor and can be found near lines:*
*13-15 - this sentence should convey something like "because the high-frequency portion of the wave spectrum integrates preceding wind conditions"*
*23   73   77   86-89   124-125   129   169   200-201-provide citation   248*

The author would like to thank the reviewer again for pointing out these minor issues. The manuscript has been revised accordingly. For the citation problem in L200-201, "The

surface currents are generally larger in coastal regions (tides) and westerlies (wind drifts) than in low-latitude open oceans", this point has become "common sense" of the oceanography to some extent and can be found in a lot of textbooks (in the sections of tides and circulation). Therefore, the author failed to find a suitable citation here. It will be nice if the reviewer can recommend one or two.